# Sub-nanosecond all-optically reconfigurable photonics in optical fibres

Kunhao Ji [1] ✉, David J. Richardson [1,2], Stefan Wabnitz [3] & Massimiliano Guasoni[1] ✉

Reconfigurable photonic systems provide a versatile platform for dynamic, on-demand control and switching. Here we introduce an all-optical platform in multimode and multicore fibres. By using a low-power probe beam and a counter-propagating control beam, we achieve dynamic control over light propagation within the fibres. This setup ensures simultaneous phase-matching of all probe-control beam four-wave mixing interactions, enabling all-optical reconfiguration of the probe modal state by tuning the control beam power. Key operations such as fully tuneable power splitting and mode conversion, core-to-core switching and combination, along with remote probe characterization, are demonstrated at the sub-nanosecond time scale. Our experimental results are supported by a theoretical model that extends to fibres with an arbitrary number of modes and cores. The implementation of these operations in a single platform underlines its versatility, a critical feature of next-generation energy-efficient photonic systems. Scaling this approach to highly nonlinear materials could underpin photonic programmable hardware for optical computing and machine learning.

The ability to manipulate light by light within optical fibres represents a pivotal advance, both for the development of new photonic technologies and the exploration of novel physical phenomena. Ground-breaking all-optical devices and applications have been developed in single-mode fibres, including optical amplifiers[1,2], signal regeneration[3], polarisation control[4,5], sensing[6] and logical operations[7].

The recent renewed interest in multimode fibres (MMFs) and multicore fibres (MCFs), driven by the need for high-speed communication systems based on space-division multiplexing (SDM)[8,9], has sparked attention to complex nonlinear multimode processes that have no counterpart in the single-mode platform, and whose comprehension is still in the early stages[10–14], paving the way for new methods of all-optical control of light.

In the framework of all-optical control, we can differentiate between self-organisation and external control. Self-organisation occurs when an intense light beam reshapes its own dynamics, owing to the substantial nonlinearity induced by its large peak power.

Beam self-cleaning[15,16], self-switching[17,18], self-coherent combination[19], and self-repolarisation processes[20,21] induced by Kerr nonlinearity fall into this category. Conversely, external control occurs when the dynamics of a probe beam are controlled by an external independent control beam through their mutual nonlinear interaction.

When both the probe and control beam are relatively intense, their nonlinear interaction may exhibit robust modal attraction[22–25] or even rejection dynamics[26], as recently demonstrated in multimode systems. In contrast, when the probe beam operates in a low-power (linear) regime, substantially different dynamics emerge, where the control beam induces a periodic optical grating inscribed in the fibre. Optically induced gratings, so far limited to bimodal systems, have been exploited to implement partial mode conversion of the probe beam[27–31].

In this work, we propose a counter-propagating probe-control beam scheme in MMFs and MCFs with arbitrary number $N$ of modes or cores. This setup allows the simultaneous phase-matching of several

[1]Optoelectronics Research Centre, University of Southampton, Southampton, United Kingdom. [2]Microsoft (Lumenisity Limited), Unit 7, The Quadrangle, Abbey Park Industrial Estate, Romsey, United Kingdom. [3]Department of Information Engineering, Electronics and Telecommunications (DIET), Sapienza University of Rome, Rome, Italy. ✉e-mail: k.ji@soton.ac.uk; m.guasoni@soton.ac.uk

interaction processes between a low-power, forward probe and an intense backward control beam (BCB), regardless of the fibre parameters, thus harnessing the full potential of multimode dynamics. By leveraging a robust setup for accurate mode coupling and mode decomposition, we provide an experimental demonstration of several compelling all-optical operations in MMFs and MCFs, which include fast and fully tuneable mode conversion and power splitting, selective core-to-core switching and combining, as well as the remote characterisation of the probe beam, as illustrated in Fig. 1.

These outcomes reveal an all-optical control mechanism for configuring the modal state and optical pathways in MMFs and MCFs, enabling novel functionalities for future smart and adaptive optical systems. This lays the groundwork for an all-optically reconfigurable photonics in optical fibres and beyond[32].

## Results

### Probe-control beam interaction

A forward probe signal and a BCB are counter-propagating in a polarisation-maintaining multimode (or multicore) fibre supporting $N$ spatial modes. Their spatio-temporal evolution is described by a system of coupled nonlinear Schrödinger equations[26] (CNLSEs, see Eq. (6) in "Methods" -Theoretical Framework). The counter-propagating setup offers fundamental key advantages with respect to a standard co-propagating configuration.

First, it enables physical separation between probe and BCB, which are launched at the opposite ends of the fibre. The physical separation allows both beams to share the same polarisation and be centred at the same wavelength $\lambda$−key conditions that define our experimental scenario. Crucially, under these circumstances, all intermodal four-wave mixing interactions between the probe and BCB are simultaneously phase-matched, regardless of the fibre parameters and beam wavelengths (see "Methods"−Theoretical Framework for details).

This simultaneous phase-matching condition is the cornerstone of our approach. It enables energy exchange across all $N$ probe modes, thereby allowing a complete reconfiguration of its modal state. The energy exchange is mediated by the BCB but without any net power transfer to the probe, which therefore preserves its total energy. Such dynamics is not attainable in conventional co-propagating systems, where the probe-BCB separation relies on differences in polarisation and/or wavelength. These constraints inherently prevent simultaneous phase-matching of all intermodal four-wave mixing interactions, typically allowing energy exchange between only two probe modes−and

only under strict conditions on fibre parameters and probe-to-BCB frequency detuning.

In addition, the probe-BCB physical separation in the counter-propagating setup allows for the implementation of remote sensing operations, hence, to investigate the properties of the fibre and/or of the probe, even when the latter is inaccessible, which is one of the applications discussed later in this work.

In a recent work[26] we analysed the case where both probe and BCB operate in a strongly nonlinear regime, which exhibits robust mode attraction or rejection states, irrespectively of the initial state of the probe. In this study, we explore a different scenario, where the probe is in the linear regime, whereas the BCB remains in a strongly nonlinear propagation regime.

The distinction between linear and nonlinear regime of the probe is governed by its total peak power $P_{pr,tot}$, and the interaction length $L_{int}$ between the probe and the BCB. In the continuous-wave (CW) case, $L_{int}$ corresponds to the full fibre length $L$, whereas for pulsed beams, it is determined by their temporal overlap. Specifically, the BCB modulates the probe over a spatial extent $L_{int} = \tau_p c$ in pulsed operation, where $\tau_p$ is the BCB pulse width and $c$ is the velocity of light in the fibre. As a rough guideline, if the number of probe nonlinear lengths $P_{pr,tot} \cdot L_{int} \cdot \gamma$ exceeds 5, where $\gamma$ is the average intermodal Kerr coefficient, then the probe operates in a strongly nonlinear regime. Conversely, if this value is below 0.5, the probe remains in a linear regime. Intermediate values between 0.5 and 5 define a transitional regime, where nonlinear effects may partially develop but do not fully govern the system dynamics. Note that a similar distinction between linear and nonlinear regime can be made for the BCB based on its number of nonlinear lengths.

The probe in linear regime leads to peculiar new dynamics, fundamentally different from mode attraction and rejection. After some algebra, we recast the CNLSEs into the following linear transformation (see Methods-System Linearization):

$$F_{out} = \mathbf{M} F_{in} \qquad (1)$$

where $F_{in}$ and $F_{out}$ are vectors of length $N$ whose entries $f_{in,n}$ and $f_{out,n}$ indicate the amplitude of the electric field of the probe mode $n$ at the input and output of the fibre, respectively, whereas $\mathbf{M}$ is a $N \times N$ matrix whose elements are defined by the BCB mode state, along with the nonlinear Kerr coefficients of the fibre and the modal propagation constants.

Besides describing the mode dynamics in MMFs and MCFs, Eq. (1) also characterises the core-to-core interaction in MCFs, following the

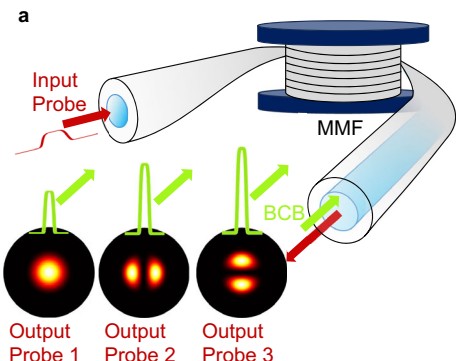

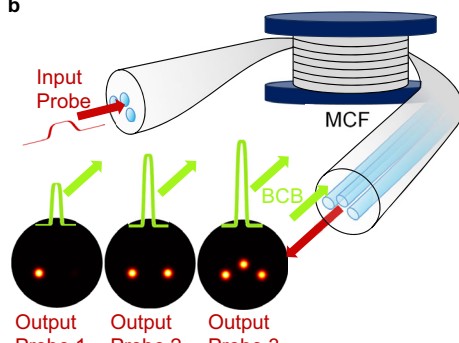

**Fig. 1 | Illustration of all-optically reconfigurable photonics in optical fibres. a** A low-power probe beam (red colour) and a high-power counter-propagating backward control beam (BCB, green colour) are injected at the two opposite ends of a multimode fibre. The BCB is coupled over a suitable combination of modes. A specific output probe on demand can be obtained by solely adjusting the BCB power. In this example, 3 different BCB intensities lead to an output probe coupled over 3 distinct fibre modes (see Output Probe 1, 2, 3). **b** Same as panel a, but in the

case of a multicore fibre with 3 cores. In this example, by tuning the BCB intensity, the output probe is either fully readdressed over a single core (Output Probe 1), or equally split over 2 (Output Probe 2) or 3 cores (Output Probe 3). The ability to manipulate the probe can be exploited to implement power splitters, mode converters and core-to-core switchers with all-optical reconfiguration at the sub-nanosecond scale.

identification of the transformation matrix $\mathbf{T}$ that maps the electric field of the MCF modes to the electric fields in the individual cores, namely:

$$F_{c-in(out)} = \mathbf{T} F_{in(out)} \tag{2}$$

where $F_{c-in}$ and $F_{c-out}$ are vectors of length $N$ whose entries $f_{c-in,n}$ and $f_{c-out,n}$ indicate the amplitude of the electric field of the probe in core $n$ at the input and output of the fibre, respectively.

Vector $F_{in}$ ($F_{out}$) describes the input (output) probe mode state, which includes information on both the input (output) mode power distribution $|F_{in}|^2$ ($|F_{out}|^2$) and the relative phase between the modes at the input (output) of the fibre. Likewise, vectors $F_{c-in}$ and $|F_{c-in}|^2$ ($F_{c-out}$ and $|F_{c-out}|^2$) represent the input (output) probe core state and power distribution, respectively. Additionally, we can define the BCB mode state and power distribution in a similar manner.

The importance of Eqs. (1) and (2) lies in their establishment of a direct relationship between the input and output probe states. Given that the matrix $\mathbf{M}$ allows reconfiguring the output probe state as a function of the input state, we call it the reconfiguration matrix. Specifically, by appropriately adjusting the BCB, we can shape the reconfiguration matrix $\mathbf{M}$ to achieve a mode or core state on demand in the output probe. In other words, we implement all-optical reconfiguration of the output probe.

### Illustration of all-optical reconfiguration in the case $N = 2$

Two simple yet insightful cases to illustrate the concept of all-optical reconfiguration of a weak probe (namely, in the linear regime) involve a multimode fibre supporting two modes and a multicore fibre with two cores in the CW regime. These cases, discussed below, will also provide useful insights for the three applications examined later.

In a multimode fibre with two modes, assuming the probe and BCB are co-polarised, the reconfiguration matrix $\mathbf{M}$ turns out to be:

$$\mathbf{M} = \begin{bmatrix} [\cos(\frac{sL}{2}) + \mathrm{i}\frac{q}{s}\sin(\frac{sL}{2})]e^{\mathrm{i}\alpha_1 L} & \mathrm{i}\frac{2r}{s}\sin(\frac{sL}{2})e^{\mathrm{i}\alpha_1 L} \\ \mathrm{i}\frac{2r}{s}\sin(\frac{sL}{2})e^{\mathrm{i}\alpha_2 L} & [\cos(\frac{sL}{2}) - \mathrm{i}\frac{q}{s}\sin(\frac{sL}{2})]e^{\mathrm{i}\alpha_2 L} \end{bmatrix} \tag{3}$$

where $q = \gamma_{11}P_{BCB,1} - \gamma_{22}P_{BCB,2}$; $r = 2\gamma_{12}(P_{BCB,1}P_{BCB,2})^{1/2}$; $s = (q^2 + 4r^2)^{1/2}$; $\alpha_1 = -\beta_1 + \frac{3}{2}\gamma_{11}P_{BCB,1} + \frac{1}{2}\gamma_{22}P_{BCB,2} + 2\gamma_{12}P_{BCB,2}$; $\alpha_2 = -\beta_2 + \frac{1}{2}\gamma_{11}P_{BCB,1} + \frac{3}{2}\gamma_{22}P_{BCB,2} + 2\gamma_{12}P_{BCB,1}$; $\gamma_{11}, \gamma_{12}, \gamma_{22}$ are nonlinear Kerr coefficients that depend on the modal spatial profiles[33]. $\beta_1$ and $\beta_2$ denote the propagation constants for mode 1 and mode 2, respectively. $P_{BCB,1}$ and $P_{BCB,2}$ represent the BCB power coupled to mode 1 and mode 2, respectively. While the BCB undergoes nonlinear phase accumulation, its powers $P_{BCB,1}$ and $P_{BCB,2}$ remain unaffected by the interaction with the weak probe (see "Methods"-System linearization).

In the absence of substantial propagation losses, dispersive effects and intermodal walk-off - conditions typically met in fibres up to a few metres long-the total instantaneous power of the probe $P_{pr,tot}$ is conserved. Starting from Eq. (1), we derive the following expressions for the instantaneous output probe power $P_{pr\,out,1} \equiv |f_{out,1}|^2$ and $P_{pr\,out,2} \equiv |f_{out,2}|^2$ coupled to mode 1 and mode 2, respectively:

$$P_{pr\,out,1} = P_{pr\,in,1} + 4(P_{pr\,in,2} - P_{pr\,in,1})\frac{r^2}{s^2}\sin^2(\frac{sL}{2}) +$$
$$+ 4\frac{r}{s}\sin(\frac{sL}{2})(P_{pr\,in,1}P_{pr\,in,2})^{\frac{1}{2}}\left[\frac{q}{s}\sin(\frac{sL}{2})\cos(\Delta\phi_{in,12})\right.$$
$$\left. - \cos(\frac{sL}{2})\sin(\Delta\phi_{in,12})\right]$$

$$P_{pr\,out,2} = P_{pr,tot} - P_{pr\,out,1} \tag{4}$$

where $\Delta\phi_{in,12}$ defines the input relative phase between the two input probe modes, $P_{pr\,in,1} \equiv |f_{in,1}|^2$ and $P_{pr\,in,2} \equiv |f_{in,2}|^2$ are the input probe powers in mode 1 and mode 2, respectively.

Equation (4) makes the concept of all-optical probe reconfiguration evident: indeed, we note that by properly setting the BCB powers $P_{BCB,1}$ and $P_{BCB,2}$, which determine the coefficients $q$, $r$ and $s$, we can control the mode power distribution of the probe at the fibre output, namely $P_{pr\,out,1}$ and $P_{pr\,out,2}$.

In the case of a multicore fibre with 2 cores, the $2 \times 2$ matrix $\mathbf{T} = [1\sqrt{2}, 1\sqrt{2}; 1\sqrt{2}, -1\sqrt{2}]$. The general solution for the output probe power coupled to each core is inherently complex. However, a particularly relevant case—yielding simpler expressions—arises when the input probe is coupled to a single core, e.g., core 1, and the BCB, with total power $P_{BCB}$, is coupled to a single mode.

In this instance the output probe power $P_{prc-out,1} \equiv |f_{c-out,1}|^2$ and $P_{prc-out,2} \equiv |f_{c-out,2}|^2$ are coupled respectively to core 1 and core 2, and computed from Eqs. (1) and (2), reads as:

$$P_{prc-out,1} = P_{pr,tot}\cos^2(\pi L/L_b + \Delta\gamma P_{BCB}L)$$
$$P_{prc-out,2} = P_{pr,tot}\sin^2(\pi L/L_b + \Delta\gamma P_{BCB}L) \tag{5}$$

where $L_b = 2\pi/|\beta_1 - \beta_2|$ is the beat-length between the two modes of the fibre having propagation constants $\beta_{1(2)}$, and $\Delta\gamma = \gamma_{11} - \gamma_{12}$.

Once again, the idea of an all-optical reconfiguration mediated by the BCB emerges. In particular, regardless of the beat-length $L_b$, the output probe power in the two cores is fully tuneable by adjusting the BCB power $P_{BCB}$.

Naturally, the concept of reconfiguration remains valid for $N > 2$ modes or cores and in the non-CW regime (see "Methods"). However, in such cases, simple analytical expressions like those in Eqs. (4) and (5) are no longer available.

### Application 1: tuneable mode manipulation

A first key application of our platform is the tuneable mode manipulation of the probe. In our experimental setup (see Fig. 2 and "Methods"-Experiments for details), the probe and BCB, centred at 1040 nm wavelength, are co-polarised and coupled at the opposite ends of the fibre under test. The probe power is fixed at a low level to ensure operation in the linear regime, while the BCB power is gradually increased to reach the high-peak power required. We used a variety of polarisation-maintaining fibres (see Supplementary Information 1 for details on their parameters) supporting 2, 3, and 6 modes at the wavelength of 1040 nm. These include a highly nonlinear fibre that relaxes substantially the power requirements on the BCB.

The results in Fig. 3 demonstrate a tuneable all-optical mode conversion in a homemade bimodal fibre, where any arbitrary power ratio between the two guided modes can be achieved by solely adjusting the BCB power. Three distinct instances are shown that highlight the extent of the precision in manipulating the probe mode distribution. Indeed, for a given input mode state of the probe, we can configure the BCB to achieve either full mode conversion of the output probe (Fig. 3d), partial mode conversion (Fig. 3e), or conversion annihilation, thus making the output probe mode distribution insensitive to the probe-BCB interaction (Fig. 3f).

Remarkably, our experimental results in Fig. 3d–f closely align with the theoretical predictions derived from the analytical solutions in Eq. (4). Note that the relative polarisation between probe and BCB may serve as an additional parameter for controlling the probe dynamics (see Supplementary Information 2). We have recorded three videos illustrating the tuneable mode manipulation dynamics in this bimodal fibre, corresponding to Fig. 3d–f (see Supplementary Movies 1–3).

Figure 4 presents additional results in three commercially available fibres: a PM1550-xp and a highly nonlinear fibre PMHN1, supporting 3 modes, along with a PM2000 supporting 6 modes.

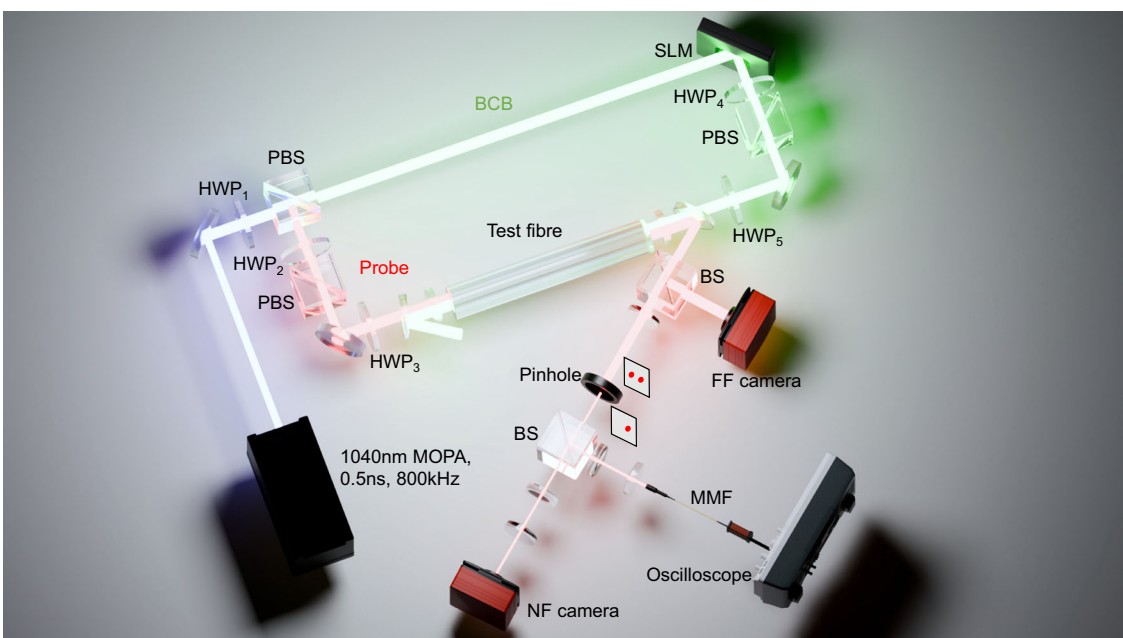

**Fig. 2 | Schematic of the experimental setup.** The input probe and BCB are split from a master oscillator power amplifier (MOPA) and coupled to the opposite ends of the test fibre. The MOPA delivers 0.5 ns pulses at a central wavelength of 1040 nm and with peak power up to 30 kW (12 W average power at 800 kHz repetition rate), therefore enabling a significant level of nonlinearity in the fibres under test. Polarisation beam splitters (PBS) and half-wave-plates (HWP$_{1-5}$) are used to tune independently the input probe and BCB power and polarisation. A near-field (NF) and a far-field (FF) camera measure the near and far field images used in our mode decomposition algorithm. The field at the output of each core of MCFs can be isolated via a pinhole and its temporal dynamic is monitored at the oscilloscope. SLM = spatial light modulator; BS = beam splitter.

Figure 4a, b illustrate the results in the PM1550-xp fibre. By adjusting the BCB mode distribution (reported at the top of each panel) we trigger and control different dynamics in the output probe. For instance, at a BCB peak power of ~11 kW (4.4 W average power), we can achieve either an equal power distribution between two of the three modes (Fig. 4a) or among all three modes (Fig. 4b).

Figure 4c, d display output probe manipulation in the PMHN1 fibre. The input probe mode state is similar in both configurations shown in Fig. 4c, d. Again, the BCB is properly adjusted to trigger different dynamics, with most of the output probe power redirected either on mode LP$_{01}$ (Fig. 4c) or LP$_{11o}$ (Fig. 4d). It is worth noting that the BCB peak power required to achieve relevant dynamics is substantially lower than in the case of the PM1550-xp, being as small as ~1 kW, which corresponds to 0.4 W average power. This is primarily due to the high nonlinearity of this fibre, resulting in significantly larger Kerr nonlinear coefficients (see Table S1 in Supplementary Information 1). This result is particularly significant as it provides experimental confirmation of the possibility to downscale the required power based on the fibre's nonlinearity. Consequently, it demonstrates the potential for a drastic reduction in energy consumption when using highly nonlinear materials, without compromising the capability for all-optical reconfiguration.

Finally, Fig. 4e, f present the results for the PM2000 fibre, which supports 6 modes. Here, the increased number of modes, combined with a lower coupling efficiency of the BCB in this fibre, reduces the effective BCB power coupled to each mode. This partially limits the extent of the probe's modal reconfiguration. This limitation could be mitigated by employing highly nonlinear fibres—such as that used in Fig. 4c, d—and by improving the coupling efficiency of the experimental setup. Despite these constraints, intriguing dynamics are still observed. In Fig. 4e, as the BCB power increases, the LP$_{01}$ mode transfers a significant portion of its power to the LP$_{11e}$ mode, while the LP$_{02}$ mode completely transfers its power to the LP$_{21e}$ mode. In Fig. 4f, at a BCB peak power of 7 kW, the LP$_{01}$ and LP$_{02}$ modes lose

approximately 8% and 6% of their initial power (measured with BCB off), which is redistributed to generate the LP$_{21e}$ and LP$_{21o}$ modes.

## Application 2: tuneable power splitting, core-to-core switching and combining

A significant feature of our setup lies in the possibility to manipulate the core-to-core energy exchange in MCFs. Note that while the mode distribution remains largely unaffected by linear coupling in the short fibres under test, core-to-core linear coupling takes place over a much shorter length scale instead. More generally, a complex interplay occurs between linear core-to-core coupling and nonlinear coupling between probe and BCB. An instructive scenario is the one previously introduced, namely, a dual-core fibre (DCF) where the input probe is coupled to a single core and the BCB is coupled to a single mode, which is described by Eq. (5).

According to the latter, when BCB is off ($P_{BCB} = 0$), the probe undergoes core-to-core energy exchange over a distance as short as $L_b$ (typically a few millimetres). Modal beat-lengths are severely affected by fibre perturbations, such as local bending and temperature fluctuations, and are therefore difficult to estimate. However, Eq. (5) highlights a crucial point. Irrespectively of the beat-length $L_b$, which may even be unknown, the output probe power in the two cores is fully tuneable by adjusting the BCB power $P_{BCB}$, enabling any arbitrary splitting ratio. Importantly, this finding is generalisable to different fibre parameters and input conditions.

Our experimental results, shown in Fig. 5a, confirm this scenario. The input probe launch condition was adjusted such that, with the BCB off, the output probe power was fully coupled to a single core. By introducing the BCB and tuning its peak power between 0 and 9 kW (average power between 0 and 3.6 W), we achieved any arbitrary power ratio X/(100 − X) between the two output cores, covering the full operational range from 100:0 to 50:50, as required for a fully tuneable optical power splitter. Similarly to the case of mode manipulation in the PMHN1 fibre reported in Fig. 4c, d, also in this case, the required BCB power could be reduced by one order of magnitude in

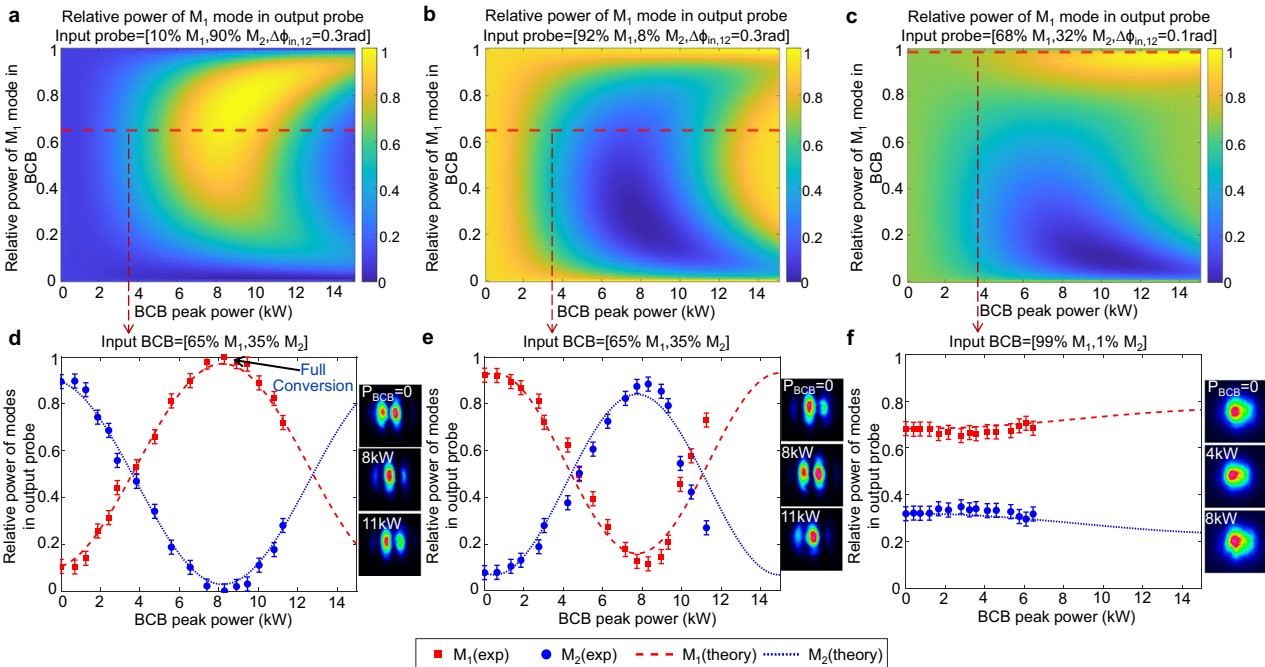

**Fig. 3 | Tuneable mode manipulation. Results in a bimodal fibre.** This fibre is 0.4 metre long and supports one even mode $M_1$ and one odd mode $M_2$ (see Supplementary Information 1). **a–c** Theoretical 2D maps of the output probe mode distribution computed from Eq. (4). The maps show the output probe power fraction coupled to mode $M_1$ versus the BCB total peak power (horizontal axis) and BCB mode distribution (vertical axis, indicating the fraction of BCB power coupled to mode $M_1$). These maps indicate how to set the BCB in order to manipulate the output probe, ensuring it reaches the desired mode distribution. The maps correspond to 3 examples with different input probe mode states, which are reported at the top of each panel. For example, in panel a the input probe mode state is characterised by 10% power on mode $M_1$, 90% on mode $M_2$, and a relative phase

$\Delta\phi_{in,12}$ between the two modes of 0.3 rad. **d–f.** Experimental (exp) and theoretical (theory) results for the same input probe mode states as panels (**a–c**), but with a fixed BCB mode distribution (indicated at the top of each panel and corresponding to the red-dashed lines in panels (**a–c**)). Arbitrary output probe mode distribution can be achieved by tuning the BCB power. Specifically, in panel (**d**), full conversion to mode $M_1$ is achieved when the BCB peak power is ~ 8 kW (3.2 W average power). In contrast, the BCB in **f** is configured such that it results in almost no variation of the output probe mode distribution. The insets in panels (**d–f**) show the far-field intensities of the output probe for different values of BCB peak power $P_{BCB}$. Error bars of ±3% are added to the measured relative power of each mode, which represents the estimated uncertainty of our mode decomposition algorithm.

highly-nonlinear fibres. This would lower the BCB average power to just a few hundred milliwatts.

Additional key applications can be envisaged and demonstrated with our platform. As shown in Fig. 5b, the power of the output probe, which in this case is relatively uniform in the 2 cores when BCB is off (power ratio core 1/core 2 = 35/65), can be combined into core 1 when the BCB peak power is set to 11 kW. Furthermore, core-to-core switching is depicted in Fig. 5c, where the output power transitions from one core to another at a BCB peak power of ~10 kW. Note that, in this case, the switching power ratios (from 15/85 to 85/15) are constrained by the available coupled BCB peak power, which is <12 kW in our experiments in the DCF. Approximately 18 kW of BCB peak power would be required for complete 0/100 to 100/0 switching (indeed 9 kW allows 100/0 to 50/50 splitting, see Fig. 5a).

The applications highlighted above can be controlled at a rapid rate through the BCB. Figure 5d, e illustrate an example of core-to-core power swapping at the sub-nanosecond time scale. The temporal evolution of the output probe power in the 2 cores, measured via an oscilloscope, is displayed. A single 0.5 ns BCB pulse shifts the core-to-core power ratio at the DCF output from 35/65, when the BCB is off, to 65/35 when the BCB peak power is 5 kW (2 W average power). The switching time is determined by the BCB pulse width. Although in our experiments the BCB pulse width is 0.5 ns, the simulations in Supplementary Information 3 indicate that the switching time could be reduced to picosecond levels. These results pave the way for the development of all-optically controlled core-to-core switchers, leading to the pioneering idea of all-optically programmable photonics. In particular, the DCF with BCB control could serve as basic unit (2 × 2

optical gate) for reconfigurable wide matrices[34,35], enabling fully optical ultrafast operations.

In this framework, exploring complex multicore systems is compelling. A single BCB could enable core-to-core switching, splitting or combining with $N > 2$ cores. These systems are more sensitive to weak variations in fibre parameters than the DCF. Our generalised solutions in Eqs. (1) and (2), which effectively describe the modal dynamics, would require precise knowledge of the relative differences among intermodal beat-lengths in order to describe the core-to-core dynamics equally well. However, these differences are susceptible to perturbations, therefore their estimation is challenging. Consequently, in our experiments we manually adjust the BCB mode state to find the optimal configuration that enables the desired control over the probe beam.

Despite these challenges, our theoretical model remains invaluable, suggesting intriguing scenarios. For instance, the simulation results in Fig. 6a indicate that, with sufficient BCB power, coherent combination or equal splitting could be achieved in a three-core-fibre (TCF). Preliminary experimental tests support the feasibility of these outcomes. Although the coupled BCB power is significantly lower than the simulated values, preventing full power rerouting in each core, nevertheless we could split the power evenly across the 3 cores (Fig. 6b), combine power from 2 cores into a single core (Fig. 6c) or swap the power among selected cores (Fig. 6d).

### Application 3: probe remote characterisation
Our counter-propagating setup could have significant applications in remote sensing, enabling the investigation of fibre or input probe features through the analysis of the output probe's response to the

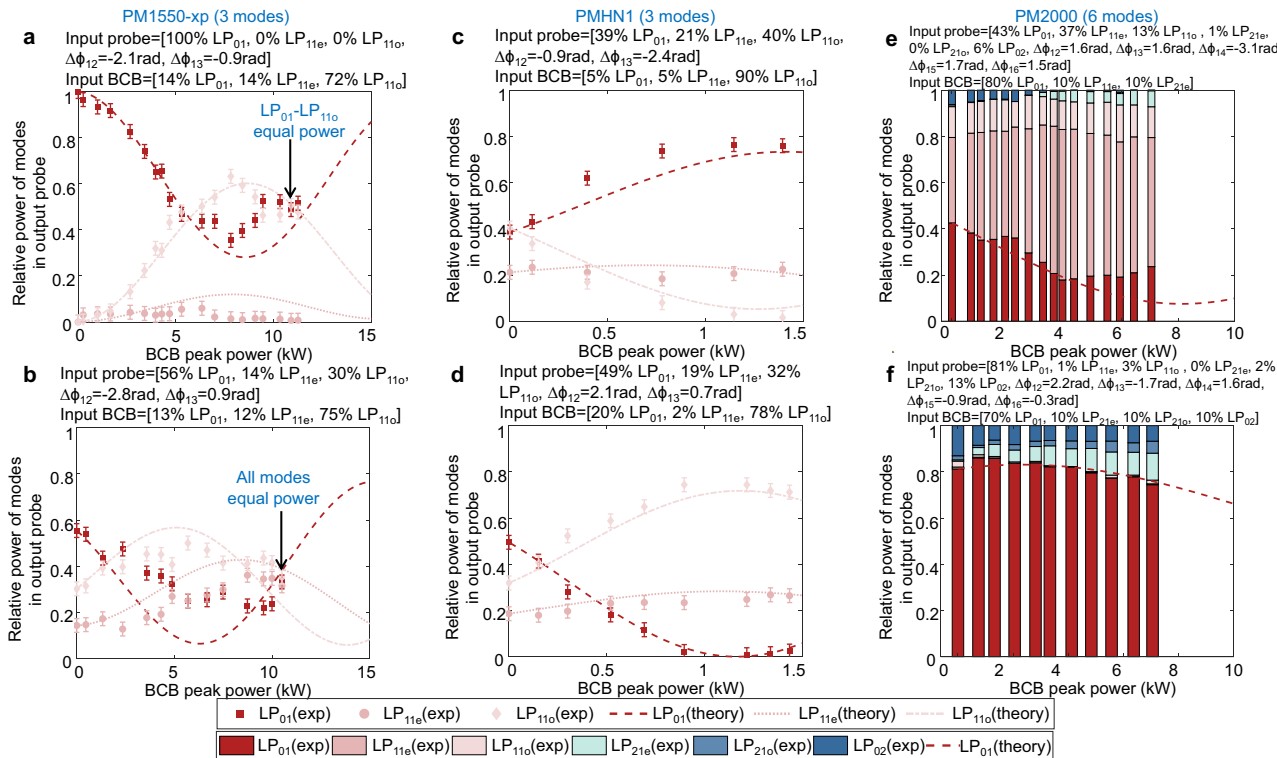

**Fig. 4 | Tuneable mode manipulation. Results in various commercially available three- and six-mode fibres.** Experimental (exp) and theoretical (theory) results are shown for different combinations of input probe and BCB mode distributions (indicated at the top of each panel) in a three-mode PM1550-xp, a three-mode PMHN1, and a six-mode PM2000 (all 0.4 m long). The six panels illustrate distinct cases of probe reconfiguration. Error bars of ±3% indicate the uncertainty in the measured relative power of each mode. Note that panels (**a**–**d**) use line plots as they involve only three modes. In panels (**e**, **f**) where six modes are involved, a bar chart is used instead to prevent excessive visual clutter. (**a**, **b**). Results in PM1550-xp fibre; (**c**, **d**). Results in PMHN1 fibre; (**e**, **f**). Results in PM2000 fibre.

BCB. For instance, consider estimating the input relative phase $\Delta\phi_{in,12}$ between two probe modes, say mode 1 and mode 2, of an MMF of length $L$.

Assuming weak mode coupling, as in few-metre long polarisation-maintaining fibres, the probe mode power distribution remains constant during propagation when the BCB is off. Conventional approaches, like the standard transfer matrix method[36], exploits the direct relationship between the output and input relative phases to estimate the latter, namely, $\Delta\phi_{in,12} = \Delta\phi_{out,12} - \Delta\phi_{acc}$, where $\Delta\phi_{out,12}$ is the output relative phase between the two modes, whereas $\Delta\phi_{acc} = \Delta\beta_{12}L$ is the accumulated phase delay due to the differential propagation constant $\Delta\beta_{12}$ between the modes. Consequently, accurate estimation of both $\Delta\phi_{out,12}$ and $\Delta\phi_{acc}$ is required, typically necessitating complex experimental setups. However, even with highly precise phase estimations, a fundamental issue remains: $\Delta\phi_{acc}$ is highly sensitive to fibre perturbations, which may result in an unreliable estimate of $\Delta\phi_{in,12}$. Maintaining accurate estimations of $\Delta\phi_{in,12}$ over time would require therefore periodic calibration or active feedback, adding further complexity.

Our platform offers an efficient solution to this problem by analysing the probe's response to the BCB. Remarkably, this approach is entirely remote, and it does not require any prior phase measurement. Indeed, as per Eq. (4), the output probe mode power distribution depends on the input relative phase $\Delta\phi_{in,12}$. Thus, to determine the latter, we computed the theoretical mode distribution for various values of relative phases and identified the optimal least-squares values that best align with the experimental data. Figure 7 illustrates three distinct cases where the input phase $\Delta\phi_{in,12}$ is successfully retrieved in a bimodal fibre, even when there is a significant power imbalance between the modes. Notably, in MCFs, once the relative phases are recovered, one may estimate through Eq. (2) the input

probe core distribution and relative phase at each core. Moreover, a similar approach could be used to estimate simultaneously both the input probe properties and unknown fibre parameters (e.g., Kerr coefficients, average linear mode coupling) through multivariate estimation analysis.

Supplementary Information 5 presents a comparison between our method and traditional transfer matrix-based techniques, highlighting the superior robustness of our approach against temperature variations and fibre bending.

## Linear vs nonlinear regime of the probe

It is useful to outline the fundamental differences between the probe dynamics in the linear regime, as considered in this study, and in the nonlinear regime[26]. These two instances exhibit substantially different behaviours. This is not surprising, as even in co-propagative classical systems—such as the simplest case of a single-mode fibre—the dynamics undergo a drastic transition from the linear regime, dominated by dispersion and polarisation effects, to the nonlinear regime, where phenomena such as solitons, rogue waves, and supercontinuum generation emerge.

As mentioned earlier, the distinction between the linear and nonlinear regimes is primarily determined by the number of nonlinear lengths. When both probe and BCB operate in a strongly nonlinear regime, the probe may undergo asymptotic attraction to or rejection of specific modal states, irrespectively of its initial state[26]. Moreover, a symmetric interaction is observed where the probe and the BCB mutually influence each other.

In contrast, when the probe operates in the linear regime (with the BCB still in a highly nonlinear regime), which is the case under investigation in this work, a fundamentally different behaviour is observed. In this instance, the interaction is highly asymmetric, with the BCB

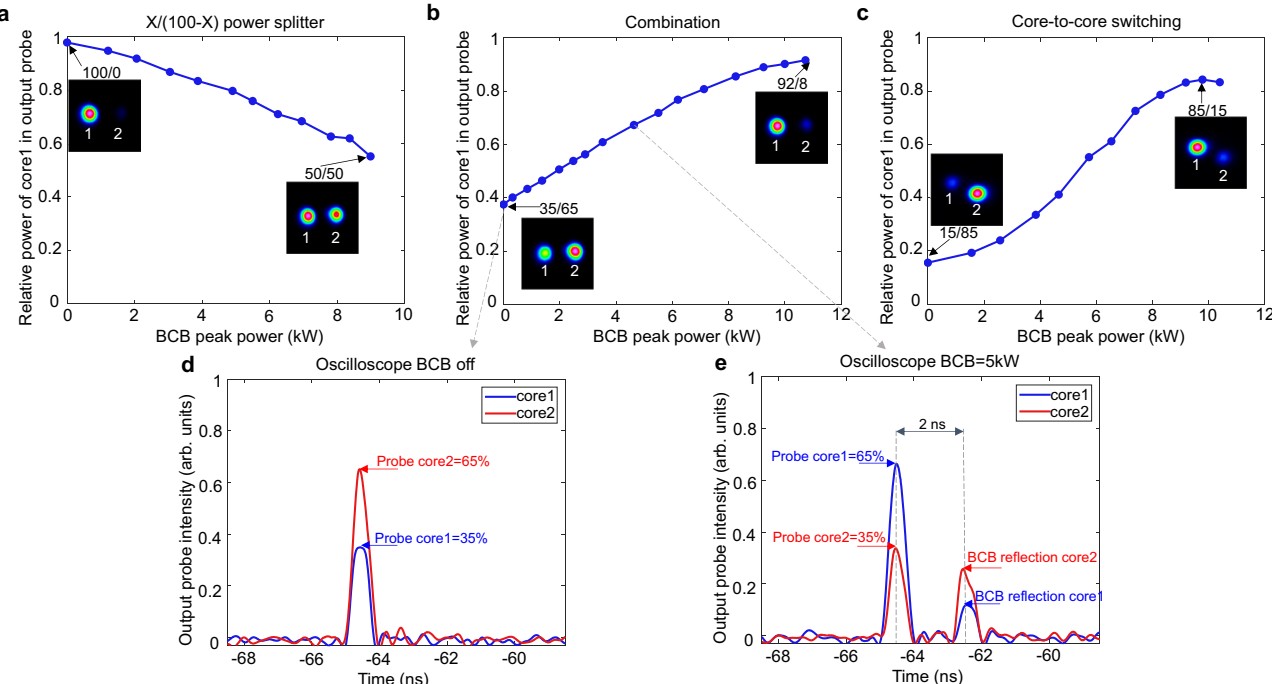

**Fig. 5 | Tuneable reconfiguration in dual-core fibre.** Three different instances are shown. The insets show the near-field intensities of the output probe at each core. **a** The input probe launch condition is optimised such that the output probe power is entirely in core 1 when the BCB is off (power ratio core1/core2 = 100/0). After having appropriately fixed the BCB mode state, we increase the BCB peak power from 0 to 9 kW (0 to 3.6 W average power). We then observe that the core-to-core power ratio of the output probe transitions gradually from 100/0 to 50/50, enabling an all-optical, fully tuneable X/(100 - X) power splitting. **b** Differently from panel a, in this case the output probe core distribution is relatively uniform when the BCB is off (power ratio core1/core2 = 35/65). The output probe is then progressively redirected into core 1 as the BCB power increases, achieving an all-optically controlled combination. At 11 kW of BCB peak power, 92% of the output probe power is in core 1 (power ratio core1/core2 = 92/8). We estimate that full combination (100/0) could be achieved at ~14 kW peak BCB power (not available). **c** In this example, the output power ratio goes from 15/85 when BCB is off to 85/15 when the BCB peak power is -10 kW. Full switching (0/100 to 100/0) could be achieved with ~18 kW BCB peak power (not available). **d** Temporal evolution of output probe power at the two cores measured by the oscilloscope when the BCB is off (power ratio core1/core2 = 35/65). **e** Temporal evolution of output probe power at the two cores measured by the oscilloscope at 5 kW BCB peak power. The power ratio shifts to 65/35. The oscilloscope also detects the BCB reflection, with the 2 ns delay corresponding to the time of flight of light in the fibre.

being only marginally affected by the probe. Moreover, the final state of the probe is strongly dependent on its initial condition, which underpins the novel applications previously illustrated.

This shift in dynamics has profound implications: rather than inducing attraction or rejection, the BCB serves as an all-optical modulator for the probe, enabling on-demand probe reconfiguration—a role traditionally fulfilled by external thermo-electronic modulators. Notably, this is achieved with a low-power probe, making the proposed applications viable for real-world implementation.

A simple numerical example in the case of a bimodal fibre allows us to clearly visualise the differences of the dynamics in the linear and nonlinear regimes. In the example shown in Fig. 8a, b, we simulate a bimodal fibre with Kerr coefficients $\gamma_{11} = \gamma_{12} = \gamma_{22} = 1/\text{W/km}$. The input probe beam is entirely coupled to mode $M_1$, while the input BCB is distributed with 60% of its power in mode $M_1$ and 40% in mode $M_2$. Their interaction length $L_{int} = 1\text{m}$. The probe beam, with a total fixed peak power $P_{pr,\ tot} = 10$ kW, operates in a highly nonlinear regime (number of nonlinear lengths $L_{int}\gamma P_{pr,\ tot} = 10$, $\gamma = 1/\text{W/km}$ being the average Kerr coefficient). As the BCB's peak power increases from 0 to 10 kW, entering itself a strongly nonlinear regime, a mode attraction process is triggered. Indeed, the output probe (Fig. 8a) tends to approach the mode state orthogonal to the input BCB, namely, ~40% on mode $M_1$ and ~60% on mode $M_2$. In turn, the output BCB (Fig. 8b) tends to approach the mode state orthogonal to the input probe, namely, all power coupled to mode $M_2$. This confirms the above-mentioned symmetric interaction between probe and BCB.

If the probe operates in the linear regime instead, the mode attraction process is not triggered. This is shown in Fig. 8c, d where the

probe peak power is now arbitrary low (here $P_{pr,tot} = 10$ mW, therefore the number of nonlinear lengths $L_{int}\gamma P_{pr,\ tot} = 10^{-5}$). In this case, the output BCB's mode composition is unaffected by the nonlinear dynamics and remains therefore unchanged, mirroring the input (Fig. 8d). Meanwhile, the output probe mode distribution exhibits a sinusoidal evolution as the BCB power increases (Fig. 8c), in line with the predictions of our theoretical model Eq. (4) and the experimental outcomes previously reported.

## Discussion

Our work presents a platform based on a counterpropagating probe-BCB setup in multimode and multicore fibres. In this setup, all the probe-BCB four-wave-mixing interactions are simultaneously phase-matched, which enables a complete reconfiguration of the probe modal state. Key operations at the sub-nanosecond time scale are demonstrated, including fully tuneable mode conversion, power splitting, core-to-core switching and combination, along with remote probe characterisation.

Unlike the system we have recently introduced in Ref. 26, this platform operates with an arbitrary weak probe. This results in fundamentally different spatiotemporal dynamics, suitable for low-power applications. Once the BCB mode state is set by the launch conditions, the BCB power can be tuned for on-demand reconfiguration of the probe.

Our experimental results are supported by a theoretical model that aligns with the experimental findings and extends to MMFs and MCFs with an arbitrary number of modes and cores. These results introduce a major shift in critical applications whose tunability

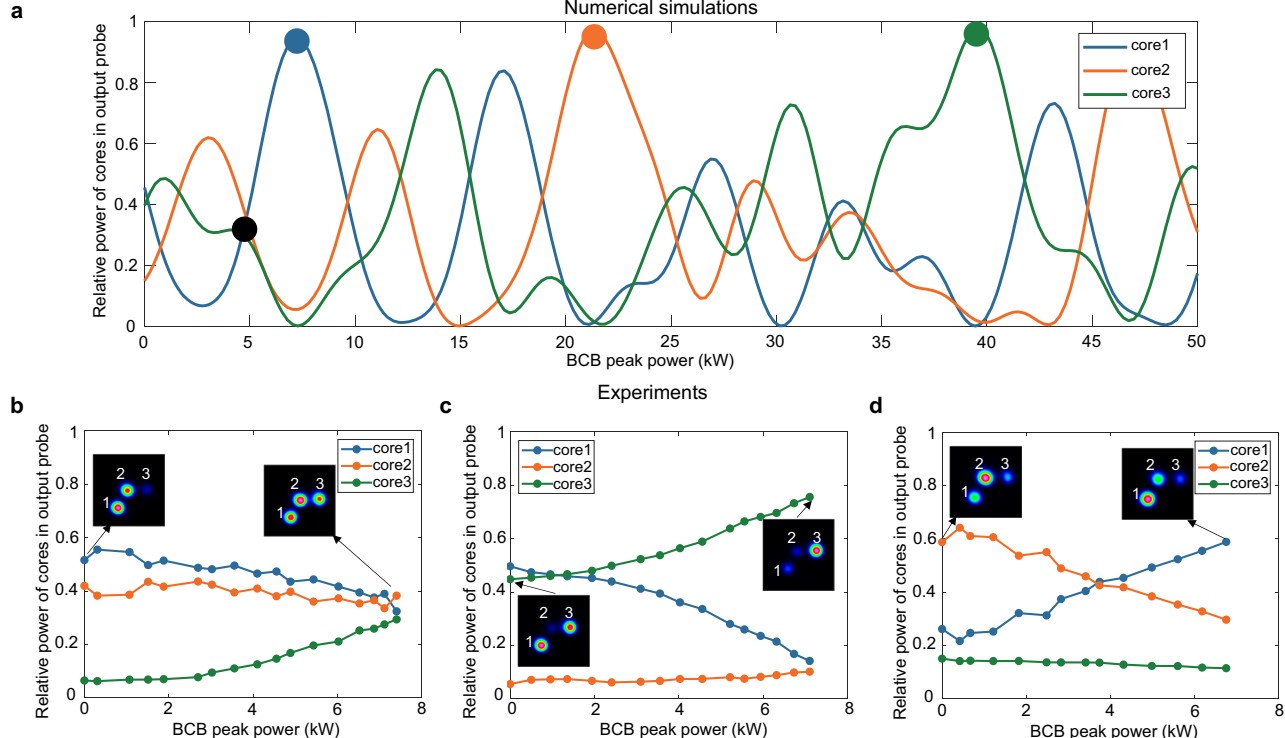

**Fig. 6 | Tuneable reconfiguration in three-core fibre.** Our ability to implement all-optical probe reconfiguration extend to fibres with more than 2 cores. This figure illustrates all-optical operations in a 0.4 m long TCF. The insets show the near-field intensities of the output probe at each core. **a** Output probe core distribution simulated via Eqs. (1) and (2), with linear and nonlinear coefficients estimated from the fibre parameters (see Supplementary Information 1). In this simulation, the BCB mode state is as follows: 5% of power in mode 1, 30% in mode 2, 65% in mode 3, and all modes in-phase. The probe power can be arbitrary low. By adjusting the BCB peak power from 0 to 50 kW we can either equalise the output probe power in the 3 cores (see black spot) or combine most of the output probe power in core 1 (blue spot), core 2(red spot) or core 3 (green spot). **b–d** Experimental results in the TCF. Each panel corresponds to different launch conditions of the input probe. In each case, the BCB is optimised to achieve relevant operations for a BCB peak power of ~7 kW (i.e., 2.8 W average power, the maximum we are able to couple into the TCF). In panel b, the output probe is almost equally split across the 3 cores. In panel c, the probe is mainly redirected to a single core (core 3). In panel d, we achieve power swapping between core 1 and core 2.

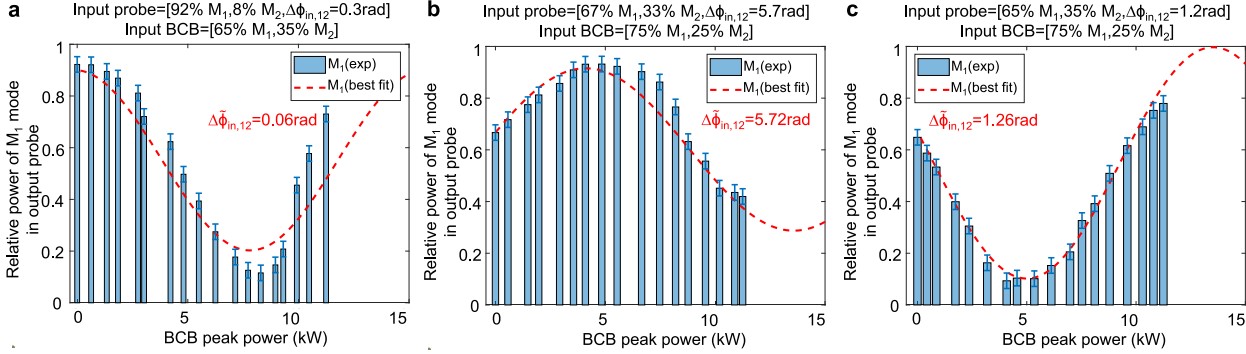

**Fig. 7 | Remote characterisation of the input probe.** Experimental results (bars) and corresponding best theoretical fits (red-dashed lines) showing the output probe power fraction coupled to mode $M_1$ versus BCB peak power in a 0.4-m long bimodal fibre (DCF, see Supplementary Information 1). Panels (**a–c**) correspond to different input probe mode states and BCB mode distributions, measured experimentally and reported on the top of each panel. The best theoretical fit is calculated from Eq. (4), assuming the same input probe and BCB relative powers and optimising the input probe relative phase to minimise the least squares difference with experimental data. Note that in all the 3 cases the estimated optimal least-squares value $\Delta\tilde{\phi}_{in,12}$ (0.06 rad, 5.72 rad, 1.26 rad in panels (**a–c**) respectively) is close to the measured $\Delta\phi_{in,12}$ (0.3 rad, 5.7 rad, 1.2 rad in panels (**a–c**) respectively). This demonstrates our ability to detect from remote the relative phase of the input probe modes by analysing the output probe response to the BCB. Note that the larger error in panel (**a**) is due to the large power imbalance among the two input probe modes (92% and 8%, respectively).

currently relies on electro-optical or thermo-optical modulation, offering a faster and more energy-efficient alternative through all-optical manipulation, a keystone for future reconfigurable optical networks and optical computing.

Among these applications, mode conversion is crucial for space-division-multiplexing[37,38]. Our platform enables not only full mode-to-mode conversion in the output probe, but more generally to achieve a tuneable combination of modes (Fig. 3 and Fig. 4). This latter capability

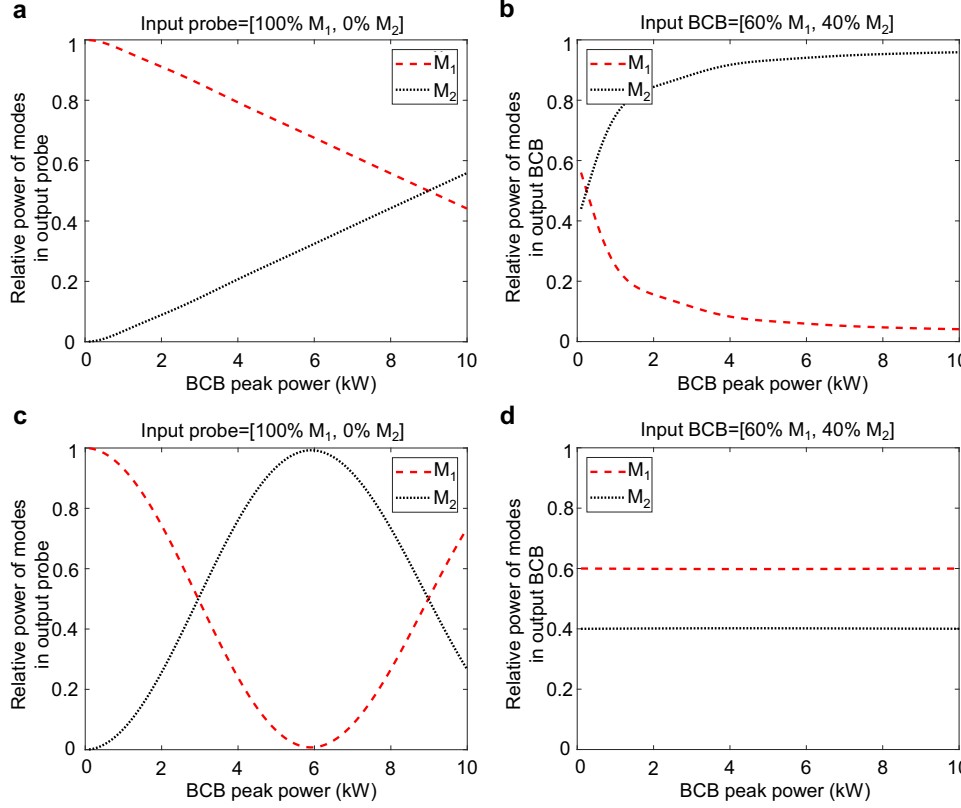

**Fig. 8 | Illustration of linear and nonlinear probe regimes (a–b).** Mode distribution of the output probe (**a**) and output BCB (**b**) versus the BCB peak power when the probe is in a strong nonlinear regime (peak power fixed to 10 kW). The output probe is asymptotically attracted to the mode state orthogonal to the input BCB, and vice versa. **c, d** Mode distribution of the output probe (**c**) and output BCB (**d**) versus the BCB peak power when the probe is in linear regime (peak power fixed to 10 mW). The output probe mode distribution oscillates sinusoidally as a function of the BCB power, whereas the BCB mode distribution is unchanged.

is essential for broadband nonlinear applications[39] and multimode interferometry[40].

Power splitting underpins power delivery, optical feedback and network access[41,42]. The ability to achieve all-optically an arbitrary splitting ratio (Fig. 5a) represents a crucial step towards real-time optimisation in time-varying scenarios such as transparent optical networks.

As for our outcomes on core-to-core switching and combining (Fig.5b, c and Fig. 6), these promise advancements in high-speed data transmission. Current switching systems are based on external devices connected to network fibres[43–45], increasing cost and complexity of the design, latency, and overall insertion losses. On the other hand, our approach suggests the feasibility of all-optical tuneable core-to-core switching directly within multicore fibres at sub-nanosecond timescale, paving the way for seamless fibre transmission through compact, all-fibre based ultrafast switchers.

Lastly, probe remote characterisation (Fig. 7) offers a novel scenario of applicability, allowing for real-time monitoring of fibre parameters or complex multimode optical signals from remote locations.

The implementation of these operations in a single platform underscores its versatility, a critical feature of next-generation photonic systems[32]. Two further points merit discussion. First, our analysis suggests that the ultimate switching time could be sub-picosecond, therefore beyond the reach of any electronic system. Moreover, scaling these results to highly nonlinear materials promises further reductions in power consumption and size. Our results in highly nonlinear fibres support this hypothesis and suggest the possibility of operating with an average optical power of just a few

hundred milliwatts, which would correspond to an electrical power consumption of less than 1 W in our experiments (wall-plug efficiency of the optical source is >50%). Notably, this power level aligns with several commercial electronic or electromechanical optical switching devices[46,47]. However, these devices exhibit relatively slow switching speeds—e.g. on the order of milliseconds for microelectromechanical systems (MEMS)—and insertion losses >1 dB. In contrast, our platform not only offers significantly higher switching speeds but is also virtually lossless, as the probe is reconfigured directly within the fibre.

We have recently demonstrated our ability to control coupling in arrays of integrated coupled waveguides[48], which represent the counterpart of MCFs on-chip. In this framework, light-by-light manipulation of the probe would add a critical degree of control for ultrafast reconfiguration at milliwatt power level. This paves the way for programmable photonics circuits[34,49,50] (hosted on MCFs, on-chip or hybrid) where basic logic blocks, like the DCF with integrated BCB, are cascaded to implement complex operations. Within this context, the ability to implement all-optically reconfigurable matrix products (see Eq. 1) may open new avenues in photonic computing and machine learning.

## Methods
### Theoretical framework
We consider two counter-propagating beams in a polarisation-maintaining multimode (or multicore) optical fibre of length $L$ supporting $N$ guided spatial modes. If the beams are co-polarised along the p-axis (p is one of the birefringence axes) and are centred at the same carrier wavelength $\lambda$, their spatio-temporal dynamic is described by the following set of coupled nonlinear Schrödinger

equations(CNLSEs)[26]:

$$\partial_z f_n + v_n^{-1}\partial_t f_n = -i\gamma_{nn}|f_n|^2 f_n + if_n\sum_{\substack{m=1}}^{N}\gamma_{nm}(\kappa|b_m|^2+2|f_m|^2)+i\kappa b_n^*\sum_{\substack{m=1\\m\neq n}}^{N}\gamma_{nm}b_m f_m$$

$$-\partial_z b_n + v_n^{-1}\partial_t b_n = -i\gamma_{nn}|b_n|^2 b_n + ib_n\sum_{\substack{m=1}}^{N}\gamma_{nm}(\kappa|f_m|^2+2|b_m|^2)+i\kappa f_n^*\sum_{\substack{m=1\\m\neq n}}^{N}\gamma_{nm}b_m f_m$$

(6)

Here $\kappa = 2$, while $f_n(z,t)$ and $b_n(z,t)$ indicate the slowly varying amplitudes of the forward and backward mode $n$, respectively. Equation (6) is completed with the boundary conditions that define the input fields, namely $f_n(0,t)$ and $b_n(L,t)$. The instantaneous amplitudes $f_n$ and $b_n$ are related to the slowly varying amplitudes through $f_n = f_n\exp\left(-i\beta_{np}z\right)$ and $b_n = b_n\exp\left(i\beta_{np}z\right)$, with $\beta_{np}(\lambda)$ the propagation constant of the p-polarised mode $n$ at wavelength $\lambda$. For the purposes of our subsequent analysis, it is useful to rewrite the relation between $f_n$ and $f_n$ in matrix form, namely $\boldsymbol{F} = \mathbf{E}_{\boldsymbol{\beta}}\mathbf{F}$, where $\boldsymbol{F}$ and $\mathbf{F}$ are $1 \times N$ vectors with elements $f_n$ and $f_n$, respectively, whereas $\mathbf{E}_{\boldsymbol{\beta}}$ is the diagonal matrix with entries $\mathbf{E}_{\boldsymbol{\beta}}[n,n] = \exp\left(-i\beta_{np}z\right)$. The coefficients $v_n$ and $\gamma_{nm}$ in Eq. (6) are the group velocity of mode $n$ and the Kerr coefficient for the nonlinear interaction between mode $n$ and $m$[33], respectively. They are computed via finite-element-method software (see Supplementary Information 1) after measuring the refractive index profile with an optical fibre analyser.

Group velocity dispersion (GVD) and higher-order dispersion terms are ignored in Eq. (6) as the corresponding characteristic lengths are substantially larger than the fibre lengths (0.4 m) used in our experiments. Similarly, detrimental nonlinear effects—including pulse reshaping and spectral broadening from self- and cross-phase modulation, Raman and Brillouin scattering, and wavelength conversion via four-wave mixing—are negligible at the peak power levels (up to a few tens of kW) and the fibre length used in our experiments. For reference, the input/output BCB temporal and spectral profiles are provided in Supplementary Information 4.

According to the normalisation of the coefficients in Eq. (6), $|f_n(z,t)|^2$ and $|b_n(z,t)|^2$ indicate the instantaneous power coupled to the forward and backward mode $n$, respectively. The total forward energy, $\int_t\sum_n|f_n|^2\partial t$, and backward energy, $\int_t\sum_n|b_n|^2\partial t$, are conserved except for propagation losses, which are negligible in the short fibres used. Similarly, when modal walk-off is negligible—which is the case for the short fibres used in our experiments—the total instantaneous powers $\sum_n|f_n|^2$ and $\sum_n|b_n|^2$ remain conserved throughout propagation.

The last summation on the right-hand-side of Eq. (6) describes the intermodal power exchange between forward and backward modes. A key feature of our counterpropagating setup is that, because the forward and backward beams are co-polarised and centred at the same carrier wavelength, each component of this summation is automatically phase-matched, irrespectively of the carrier wavelength and the fibre parameters. Consequently, a nonlinear dynamic is triggered where all modes can simultaneously exchange energy, rather than just a single pair of phase-matched modes, as typically occurs in co-propagating setups.

If forward and backward beams are orthogonally polarised along different birefringence axes, the nonlinear intermodal interaction is reduced by a factor of 1/3 ($\kappa = 2/3$ in Eq. (6)), and each component of the last summation is subject to a polarisation phase-mismatch $\Delta\beta = \beta_{nx}(\lambda) - \beta_{ny}(\lambda) + \beta_{my}(\lambda) - \beta_{mx}(\lambda)$. However, in the fibres under test, this phase mismatch barely impacts the dynamic, since the corresponding beat length $2\pi/\Delta\beta$ is typically larger than the interaction length $L_{int}$ between forward and backward beams. The latter equals the fibre length in the continuous-wave (CW) case, while reads $L_{int} = \tau_p c$ in pulsed operation, where $\tau_p$ is the pulse width of the forward and

backward beams and $c$ is the velocity of light in the fibre. Similarly, if forward and backward beams are centred at different carrier wavelengths $\lambda_f$ and $\lambda_b$, the induced phase-mismatch is negligible whenever the detuning $\Delta\lambda = |\lambda_f - \lambda_b| << \lambda_0^2/(cL_{in}|v_n^{-1} - v_m^{-1}|)$ with $\lambda_O = (\lambda_f + \lambda_b)/2$[26]. This enables tuning of the wavelength selectivity for applications such as core-to-core switching, which occurs only when the probe wavelength is sufficiently close to the BCB wavelength ($\Delta\lambda < 10$ nm in the fibres under test).

## System linearization
In the following, in accordance with the notation used in the manuscript, we indicate the forward and backward beam with probe and BCB, respectively. Let us consider the CW case in which the probe is a signal with low power. Equation (6) is reduced to Eqs. (7) and (8) by using a perturbation approach where the less significant nonlinear terms are ignored along with time-varying terms ($\partial_t f_n$ and $\partial_t b_n$):

$$\partial_z f_n = +if_n\sum_{\substack{m=1}}^{N}\gamma_{nm}\kappa|b_m|^2 + i\kappa b_n^*\sum_{\substack{m=1\\m\neq n}}^{N}\gamma_{nm}b_m f_m$$

(7)

$$-\partial_z b_n = i\theta_n b_n$$

(8)

We note that the first summation on the right-hand side of Eq. (7) represents the intermodal cross-phase modulation terms between the BCB modes and the probe modes, which are responsible for modulating the phase of the latter.

The second summation, instead, accounts for the intermodal four-wave mixing terms between the BCB and the probe, leading to the exchange of photons between probe modes. Importantly, the BCB does not transfer net power to the probe. Indeed, as previously mentioned, its total energy remains conserved, aside from propagation losses. However, the BCB acts as an intermediary, enabling energy redistribution among the probe modes and thereby a complete reconfiguration of its modal state.

Here $\theta_n = -\gamma_{nn}|b_n|^2 + \sum_{m=1}^{N}2\gamma_{nm}|b_m|^2$ plays the role of a nonlinear phase shift induced by self-phase and cross-phase modulation. The solution for the BCB mode $n$ reads as $b_n(z) = b_n(0)\exp(-i\theta_n z)$, therefore its amplitude is preserved in propagation, except for the nonlinear phase variation. We insert this solution in Eq. (7) and we use the transformation $f_n = \hat{f}_n\exp(i\theta_n z)$. This latter transformation can be recast in matrix form as $\mathbf{F} = \mathbf{E}_{\boldsymbol{\theta}}\hat{\mathbf{F}}$, where $\hat{\mathbf{F}}$ is the $1 \times N$ vector with elements $\hat{f}_n$ and $\mathbf{E}_{\boldsymbol{\theta}}$ is the diagonal matrix whose entry $\mathbf{E}_{\boldsymbol{\theta}}[n,n] = \exp(i\theta_n z)$. We finally obtain a system of linear differential equations (LDE) for $\hat{f}_n$ that can be written as $\partial_z\hat{\mathbf{F}} = i\mathbf{A}\hat{\mathbf{F}}$, where $\mathbf{A}$ is the $N\times N$ matrix whose diagonal elements $\mathbf{A}[n,n] = -\theta_n + \kappa\sum_{m=1}^{N}\gamma_{nm}|b_m|^2$, and $\mathbf{A}[n,m] = \kappa\gamma_{nm}b_m(0)b_n(0)^*$ for $n\neq m$. The matrix $\mathbf{A}$ stores therefore the information on the BCB mode state. The solution to the above-mentioned LDE system is readily found by eigenvector decomposition of matrix $\mathbf{A}$, namely $\hat{\mathbf{F}}(L) = \mathbf{V}\exp(i\mathbf{\Lambda}L)\mathbf{V}^{-1}\hat{\mathbf{F}}(0)$, where $\mathbf{V}$ and $\mathbf{\Lambda}$ are the matrices of eigenvectors and eigenvalues of $\mathbf{A}$, respectively. Now, by making use of the relations previously introduced, namely $\boldsymbol{F} = \mathbf{E}_{\boldsymbol{\beta}}\mathbf{F}$ and $\mathbf{F} = \mathbf{E}_{\boldsymbol{\theta}}\hat{\mathbf{F}}$, we derive the solution $\boldsymbol{F}(L) = \mathbf{M}\boldsymbol{F}(0)$ previously indicated as Eq. (1), where $\mathbf{M} = \mathbf{E}_{\boldsymbol{\beta}}\mathbf{E}_{\boldsymbol{\theta}}\mathbf{V}\exp(i\mathbf{\Lambda}L)\mathbf{V}^{-1}$ (with $\mathbf{E}_{\boldsymbol{\beta}}$ and $\mathbf{E}_{\boldsymbol{\theta}}$ computed in $z = L$), while $\boldsymbol{F}(0) \equiv \boldsymbol{F_{in}}$ and $\boldsymbol{F}(L) \equiv \boldsymbol{F_{out}}$ are the input and output probe mode state, respectively.

The above-mentioned solution is generally applicable to any multimode fibre system, including coupled multicore fibres. In the latter case, it is useful to derive a relationship between the field in the individual cores of the fibre. We proceed by using a couple mode theory approach, where the modes of the multicore fibre are

approximated as a linear combination of the fields in the cores, namely, $F_c = TF$, where $T$ is a transformation matrix and $F_c$ is the 1 x $N$ vector whose element $f_{c,n}$ indicates the field in the core $n$. In the simplest case of a DCF with single-mode cores, the two guided modes are well approximated as the in-phase and anti-phase sum of the fields in the cores, therefore $T = [1;1;1, -1]/\sqrt{2}$. In general, the unitary $T$ matrix strictly depends on the core-to-core arrangement. In the case of the TCF under test (corresponding results are illustrated in Fig. 6b–d), where the cores are arranged at the vertices of an isosceles triangle with 30-deg base angle and ~16.5 μm base, we have $T = \left[ \sqrt{2}, 0, \sqrt{2}; 1, \sqrt{2}, -1; 1, -\sqrt{2}, -1 \right]/2$.

When the probe is in linear regime, the solution of the full CNLSEs Eq. (6) yields the same results as the simplified system Eq. (7) and the analytical formulas Eqs. (1) and (2), confirming the validity of our model. The advantage of using Eqs. (1) and (2) is that they directly provide the probe mode/core state as a function of matrices $M$ and $T$, eliminating the need for propagation codes. Notably, Eqs. (1) and (2) allow identifying the optimal matrix $M$, and then the related optimal BCB mode state, to implement the all-optical applications introduced in this work.

Note that, in the general non-CW case, where $L_{int} < L$, the theoretical analysis proceeds as follows. We consider a fibre section of length $L_{int}$ where the probe and BCB interact, governed by the same CW-model outlined above. Beyond this, in the remaining fibre sections where no interaction occurs, we effectively assume the BCB is off. The overall solution is obtained by solving each region separately and enforcing continuity at their interface. This approach remains valid as long as L is sufficiently short to prevent excessive modal walk-off.

### Relation between probe and BCB mode/core states and mode/core power distribution

The matrix $A$ can be decomposed as $A = E^*_{\angle B_0} A' E_{\angle B_0}$, where $E_{\angle B_0}$ is the diagonal matrix whose entry $E_{\angle B_0}[n,n] = \exp(-i \arg(b_n(0)))$ identifies the phase of the BCB mode $n$ in z = 0, and $A'$ is the matrix created from $A$ by replacing the non-diagonal entries $\kappa \gamma_{nm} b_m(0) b_n(0)^*$ with the corresponding magnitude $\kappa \gamma_{nm} |b_m(0)||b_n(0)|$. Matrices $A$ and $A'$ are therefore equivalent except for the phase information of the BCB, which is missing in $A'$.

By exploiting the above-mentioned decomposition, the relation $\partial z \hat{F} = iA\hat{F}$ can be rewritten as $\partial z \left( E_{\angle B_0} \hat{F} \right) = iA' \left( E_{\angle B_0} \hat{F} \right)$, meaning that the dynamics of the transformed vector $E_{B_0} \hat{F}$ depends solely on the modified matrix $A'$. Since $\left| E_{\angle B_0} \hat{F} \right| = |\hat{F}| = |F|$, we conclude that the probe mode power distribution $|F|$ is fully determined by $A'$, rather than $A$. In other words, the output probe mode power distribution only depends on the BCB mode power distribution (that is preserved in propagation and is fixed by the launch conditions), but not on the BCB mode relative phases. This is not generally true for the output probe core power distribution, which depends instead on the full BCB mode state.

### Experiments

In our experiments, the BCB operates in a pulsed configuration rather than as a CW, which enhances the peak power and, consequently, the system nonlinearity. The probe could in principle operate in the CW regime with indefinitely low power. In practice, the probe-to-BCB power imbalance in our experiments is ~1:20. Indeed, a lower probe power would result in a weak signal-to-noise ratio, thus degrading the image quality, and preventing an accurate mode decomposition of the probe. Consequently, both BCB and probe are pulsed in our experiments (which does not change the main outcomes, see Supplementary Information 3). Specifically, 0.5 ns-pulsed probe and BCB are generated by splitting the beam from an in-house built linearly polarised ytterbium master oscillator power amplifier having central wavelength $\lambda c = 1040$ nm and a repetition rate of 800 kHz[51]. The probe and BCB are then injected at the two opposite ends of the fibres under test. Five

distinct fibres are employed (see Supplementary Information 1): a polarisation-maintaining (PM) few-mode fibre (PM1550-xp from Thorlabs) supporting 3 guided modes at λc; a highly nonlinear PM few-mode fibre (PMHN1 from Thorlabs) supporting 3 guided modes at λc; a PM few-mode fibre (PM2000 from Thorlabs) supporting 6 guided modes at λc; and then a homemade dual core fibre (DCF) and three-core fibre (TCF) supporting respectively 2 and 3 guided modes at λc. The short fibre length (40 cm) used in our experiments prevents the onset of detrimental nonlinear effects, such as Raman and Brillouin scattering, self- and cross-phase modulation-induced pulse reshaping, and four-wave mixing-induced wavelength conversion.

The input power and polarisation of probe and BCB are controlled with a proper combination of polarisation beam splitters and half-wave-plates (HWP$_2$ to HWP$_5$ in Fig. 2). By adjusting the phase pattern displayed on the screen of a spatial light modulator, we control the mode state of the input BCB, namely, its power distribution and relative phase over the fibre modes. A spatial phase plate is used to excite an arbitrary combination of modes at the probe input end for the PM1550-xp and PMHN1, while the input probe is selectively coupled into a single core to excite a combination of modes in the DCF and TCF.

The test fibre at the BCB input end is cleaved with an angle of 8-deg to eliminate back reflection of the BCB, whereas the probe input end is perpendicularly cleaved to ensure high-quality mode excitation. The output probe is sampled using a wedge with an incident beam angle of ~10 deg, ensuring that the sampled beam preserves the output probe polarisation. The near-field and far-field intensity profiles of probe and BCB are measured with infra-red cameras, with the output probe profiles corrected by subtracting the BCB reflection from the flat-cleaved fibre end. Mode decomposition of the probe and BCB is then implemented based on the measured intensity profiles. Specifically, a reconstructed spatial distribution is generated by numerically determining the mode state through an iterative process, where the Stochastic Parallel Gradient Descent algorithm[52] is successfully applied. The reconstructed distribution typically exhibits a correlation as high as 99%[53–55] with the measured spatial profile, which confirms the effectiveness of the mode decomposition method. The reconstructed mode distributions are compared with the measurements at varying BCB peak powers for the results presented in Fig. 3d–f (see Supplementary Movies 1–3).

In the MCFs under test, the power in each individual core is measured by integrating the intensities within the core areas in the near-field intensity profiles. To analyse the temporal evolution of core-to-core power switching, the output probe pulses from each core are characterised by an oscilloscope. As shown in Fig. 2, the output probe is imaged at the pinhole position via a pair of lenses (focal lengths = 13.86 mm and 500 mm, providing a magnification factor of ~36x). With a clear aperture of ~200 μm, the pinhole can effectively filter out the beam from a single core. The filtered output probe is then coupled through a telescope into a multimode fibre connected to the oscilloscope, with a replica imaged onto the camera using another telescope. Due to the flat cleave at the input-probe fibre end, the BCB reflection at this facet propagates in the same direction of the probe and can then also be measured (see BCB reflection in Fig. 5e). However, the BCB reflected pulses are separated from the output probe pulses due to the differential travelling path length, with a delay essentially determined by the fibre length (~2 ns in Fig. 5e).

## Data availability

The data are available at ref. 56.

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

## Acknowledgements

M.G. acknowledges funding from the European Research Council under the H2020 Programme (ERC Starting Grant No. 802682, MODES project), from the UK Engineering and Physical Sciences Research Council (EP/T019441/1), and from the British Council (UK-India Education and Research Initiative IND/CONT/G/23-24/36); K.J. acknowledges funding from China Scholarship Council (202006840003) and from the UK Engineering and Physical Sciences Research Council (EP/X040569/1); D.J.R. acknowledges funding from the UK Engineering and Physical Sciences Research Council (EP/P030181/1); S.W. acknowledges funding from the European Research Council under the H2020 Programme (ERC Advanced Grant No.740355, STEM project) and from European Union-Next Generation EU (PE00000001, RESTART). The authors acknowledge the use of freely available 3D components from Optical Components V1 by Ryo Mizuta Graphics (available at https://ryomizutagraphics.gumroad.com/l/OpticalComponentsV1), which were adapted and rendered in Blender for Fig. 2 in this work.

## Author contributions

K.J. performed all the experiments reported in this work and performed the numerical simulations with M.G.; D.J.R. provided support for the experimental work and contributed to the interpretation of the results; S.W. provided support for the theoretical work and contributed to the interpretation of the results; M.G. additionally conceived the research idea, supervised the project, developed the theoretical results and wrote the manuscript with feedback from all the authors.

## Competing interests

The authors decale no competing interests.
