## [Transparent Peer Review file · Nature Communications]

Sub-nanosecond all-optically reconfigurable photonics in optical fibres

Corresponding Author: Mr Kunhao Ji

Version 0:

Reviewer comments:

Reviewer #1

(Remarks to the Author)

This work is an extension of the findings of the authors' 2023 publication in Nature Communications, where they demonstrated that a backpropagated high-power pump can manipulate the mode/core distributions of a probe signal. The key difference between the two studies is that, while the earlier work involved both the pump and probe operating at high power, the current paper explores a scenario where the probe operates at low power. The authors also observe that the pump-probe interaction behavior differs significantly in the low-power probe case.

Despite these differences, I would still consider this work an extension of the prior study for the following reasons. The theoretical framework is the same, based on coupled Schrödinger equations. The experimental setup and configurations are very similar to those in the earlier study. More importantly, while the paper reports some interesting phenomena, it did not make sufficient progress toward practical applications compared to the previous work. Although a few additional applications are suggested, the demonstrated results remain far from practical relevance. For example:

1. Only two modes are demonstrated.
2. The required optical power is extremely high, reaching tens of kW.
3. The probe signals used are pulsed. To better illustrate the practical value of the findings, experiments using actual communication signals would have been more compelling.
4. The potential for ultrahigh-speed switching, highlighted in the paper title, is certainly interesting. However, the authors provide no experimental demonstration of this capability.

Moreover, the quality of the paper's presentation needs improvement. The theory section, "Probe-Control Beam Interaction" gives very limited useful information. Important details to understand the working conditions of the system—such as the power levels and the wavelength relationship between the pump and probe—are not mentioned. Additionally, the method of how to change the pump to control the matrix is not explained. Readers must frequently refer to supplementary information or the experimental results to get more information.

In summary, while the findings in this paper are interesting, the experimental demonstrations and overall presentation quality fall short of the standards expected of Nature Communications. My main concern is the lack of significant experimental progress toward practical applications, compared to the authors' previous work.

Reviewer #2

(Remarks to the Author)

I have carefully reviewed the paper titled "Sub-nanosecond All-Optically Reconfigurable Photonics in Optical Fibers" and identified several critical issues that need to be addressed and revised by the authors before the manuscript can be considered for publication.

Major Issues:

1. The concept of nonlinear mode conversion in multimode optical fibers is not new and has already been explored in previous works. Notable references include:

o J. Xu, G. S. D. Gordon, T. Wilkinson, C. Peucheret, "Experimental observation of non-linear mode conversion in few-mode fiber," CLEO, 2015, paper SM2L.3.

o R.-J. Essiambre et al., "Experimental Observation of Inter-Modal Cross-Phase Modulation in Few-Mode Fibers," IEEE Photonics Technology Letters, vol. 25, no. 6, pp. 535-538, March 15, 2013.

o Dimitar I. Kroushkov, Georg Rademacher, and Klaus Petermann, "Cross mode modulation in multimode fibers," Opt. Lett., 38, 1642-1644 (2013).

The manuscript must clearly highlight the novelty of the presented work and how it advances beyond these existing studies.

2. The manuscript does not clearly explain the underlying physics of nonlinear mode and core conversion in the backward-pumping configuration. The authors must explicitly state whether the nonlinear conversion is attributed to intermodal cross-phase modulation, four-wave mixing, or another nonlinear interaction. What specific nonlinear processes are involved? Similarly, the mechanisms behind nonlinear core conversion in multicore fibers require further clarification. Is phase matching simultaneously achieved for multiple interaction processes? More detailed explanations are essential.

3. The required pump power for nonlinear conversion is reported to exceed 7 kW. This raises concerns about nonlinear pulse propagation effects. The authors should discuss the implications of such high power levels in greater detail, including potential limitations and how they mitigate detrimental nonlinear effects.

4. The manuscript does not address how the fiber length impacts nonlinear conversion efficiency. This is an important parameter that needs to be investigated and discussed.

5. The title should clearly specify that the work involves multimode or multicore optical fibers. Otherwise, it risks appearing redundant with the extensive body of research on all-optical processing in single-mode fibers over the past four decades.

Minor Issues:

1. Figures: The resolution of the figures needs improvement to ensure clarity and readability.

2. Terminology on Page 163:

o The phrase on page 163 "PBCB is coupled to a single mode, say mode 1" is unclear. Do you mean a single core ?

Reviewer #3

(Remarks to the Author)

In this manuscript, effort has been made towards all-optically dynamical manipulation over light within optical fiber. Besides, with a low-power probe beam and a high-power counter-propagating control beam, the article expands the application of light-light modulation from single-mode fiber (SMF) to few-mode fiber (FMF) and multi-core fiber (MCF). Experiments including mode conversion, core-to-core tunable power reconfiguration and probe remote characterization are demonstrated to underline its versatile feature and its sub-nanosecond modulation time scale. To some extent, the setups do made contributions towards reconfigurable optical networks and optical computing, I still find this work not solid enough to match the high standard of Nature Communications in terms of theory and application. Therefore, I regret to tell that I cannot recommend the publication to Nature Communications. My perspectives are as follows:

1. This manuscript does not seem to differ significantly from the previous work mentioned in the Ref [1]. The input probe light has been changed from a high-power beam that induces strong nonlinearity to a low-power beam, and lead to new dynamics. However, there is no further demonstration of a deeper exploration into the complex nonlinear multimode processes during this process.

2. The manuscript demonstrates the tunable conversion between modes M1 and M2, as well as core-to-core power conversion in a three-core fiber (TCF). However, these experiments do not adequately demonstrate the universality of the proposed all-optical modulation across arbitrary scenarios. In other words, the conversion involving only two modes (and the transitions among TCF) is too fundamental to showcase the superiority of this method. In my opinion, exploring conversions among a greater number of modes could be beneficial to address more complex application scenarios.

3. I have concerns regarding the title "Sub-nanosecond all-optically reconfigurable photonics in optical fibers." The conversion time mentioned in line 192 for the multi-core fiber is not equivalent to the overall response time of the system when considering nonlinear effects. I believe this should be clearly articulated in the title to avoid ambiguity.

4. The formulation in line 95 is unclear. While T is described as the transmission matrix that connects the input and output ends, the wording in the manuscript may lead to confusion.

5. In Application 3, given that the transfer function is known, it is relatively straightforward to deduce the relationship with the input parameters through the far-field characteristics of the output. In my opinion, comparing the proposed approach with other conventional methods to demonstrate its superior inference accuracy would enhance the manuscript's credibility.

References

[1] Ji, K. H. et al. Mode attraction, rejection and control in nonlinear multimode optics. Nat. Commun. 14, 7704 (2023).

Reviewer #4

(Remarks to the Author)

The proposed novel all-optical platform in multimode and multicore fibres in the article is interesting, but it also faces many challenges and problems:

1)The probe power is weak (linear regime), whereas the BCB is in a strongly nonlinear propagation regime. This leads to peculiar new dynamics, fundamentally different from mode attraction and rejection. The power range of the probe and BCB

should be discussed to better distinguish different working dynamics (mode attraction and rejection, all-optical reconfiguration).

2) Although arbitrary output probe mode distribution can be achieved by tuning the BCB power in Fig. 3, we would like to know more about the purity of the pattern and how it varies with CBC.

3) Why was the fibre chosen to be 0.4 meter long?

4) A single 0.5 ns BCB pulse shifts the core-to-core power ratio at the DCF output from 35/65, when the BCB is 192 off, to 65/35 when the BCB peak power is 5 kW. This relative change is only 30%, how can it reach the application level of full light control, and the CBC power is also particularly high, so the application prospects seem very limited.

5) The larger error in panel a of Fig. 6 is due to the large power imbalance among the two input probe modes (92% and 8%, respectively). This means that if the power difference is too large, the error will be very large. Can this power range be preliminarily defined?

6) The most fatal problem with the method proposed in the article is that optical fibers are particularly sensitive to changes in the external environment. How to overcome this challenge should be discussed.

Version 2:

Reviewer comments:

Reviewer #1

(Remarks to the Author)

Reviewer #2

(Remarks to the Author)

I have carefully reviewed the authors' responses to the referees' comments and analyzed the revised manuscript in detail. The authors have addressed most of my initial concerns, and the conclusions are now better supported by additional data and expanded discussion. While some weaknesses remain, I believe the manuscript has improved significantly and is suitable for publication in Nature Communications.

Reviewer #3

(Remarks to the Author)

The authors have revised the manuscript, and addressed the issues I have raised before. Now I can accept it.

Reviewer #4

(Remarks to the Author)

The author's revisions and responses are both very objective, reinforcing the noteworthy results. It represents a paradigm shift in all-optical control, moving beyond conventional electro-optical methods. The combination of speed (sub-ns), energy efficiency (mW-level in highly nonlinear fibers), and multi-functionality (switching/splitting/remote sensing) aligns perfectly with next-gen photonics needs. But I hope the authors can provide a limit prediction, which is the maximum number of modes or fiber cores that this method can handle simultaneously, although new experiments conducted in a six-mode fibre was presented. By the way, the title of the article is a little bit too grand, as essentially referring to the manipulation of light

Response to reviewers' comments:

Reviewer #1:

This work is an extension of the findings of the authors' 2023 publication in Nature Communications, where they demonstrated that a backpropagated high-power pump can manipulate the mode/core distributions of a probe signal. The key difference between the two studies is that, while the earlier work involved both the pump and probe operating at high power, the current paper explores a scenario where the probe operates at low power. The authors also observe that the pump-probe interaction behavior differs significantly in the low-power probe case.

1. Despite these differences, I would still consider this work an extension of the prior study for the following reasons: The theoretical framework is the same, based on coupled Schrödinger equations. The experimental setup and configurations are very similar to those in the earlier study.

[1] Ji, K. H. et al. Mode attraction, rejection and control in nonlinear multimode optics. Nat. Commun. 14, 7704 (2023).

We respectfully disagree with this assessment. We acknowledge that this might be due to our inability to clearly convey the differences between our work and Ref. [1], previously outlined in the former Supplementary Information 4. We take this opportunity to outline these distinctions. While it is true that the experimental setup and underlying equations are similar, this does not imply that the present study is a mere extension. A useful analogy can be drawn with single-mode optical fibres: given the same equations and setup, the observed dynamics drastically change depending on whether the input light is in the low-power regime (dominated by dispersion and polarization effects) or in the high-power regime (leading to phenomena such as solitons, rogue waves, or supercontinuum generation, to name a few).

Similarly, in our previous work [1], where the probe was at high-power, we observed a symmetric interaction where the probe and the BCB mutually influenced each other, leading to soliton-like (mode attraction) or anti-soliton-like (mode rejection) dynamics. Importantly, this process occurred independently of the initial probe state. In contrast, the present study reveals a fundamentally different behaviour. First, the interaction is highly asymmetric, with the BCB being only marginally affected by the probe. Second, the final state of the probe is strongly dependent on its initial condition, which underpins the novel applications proposed.

This shift in dynamics has profound implications: rather than acting as an attractor or rejector, the BCB now serves as an *all-optical modulator for the probe*, enabling on-demand probe reconfiguration directly within the fibre—a role traditionally fulfilled by external thermo-electronic modulators.

Also from a theoretical perspective, the present work departs significantly from [1]. While in [1] the dynamics was fully characterized by a correlation function, here we introduce a novel framework

based on the reconfiguration matrix M . This allows us, for the first time, to derive the conditions required to tune the BCB and achieve the desired probe output. More broadly, the ability to reconfigure all-optically matrix M enables fast reconfigurable matrix products (see Equation 1), a functionality with profound implications for all-optical computing and photonic hardware for machine learning if implemented on chip. In conclusion, this work fully realizes the concept of all-optical reconfiguration of the modal state. Notably, this is achieved with a low-power probe, making the proposed applications viable for real-world implementation.

To ensure these fundamental differences are fully appreciated, we have created an entirely new Section: ‘Linear vs nonlinear regime of the probe’ (line 392-438), which incorporates the aforementioned points along with a new figure (Fig.8) and the content previously reported in the former Supplementary Information 4.

2. More importantly, while the paper reports some interesting phenomena, it did not make sufficient progress toward practical applications compared to the previous work. Although a few additional applications are suggested, the demonstrated results remain far from practical relevance. For example:

Only two modes are demonstrated.

We note that in the previous version of the manuscript, we had already discussed in the former Supplementary Information 2 the results obtained in three-mode fibres. However, we recognize that presenting them in the main manuscript strengthens the clarity and impact of our findings.

In the revised version, we have introduced a new Figure 4, along with corresponding comments (line 225-259), where we present both earlier results in three-mode fibres and new experiments conducted in a six-mode fibre. Overall, we showcase several distinct cases that demonstrate our ability to manipulate the probe’s modal state on demand, even in complex scenarios.

3. The required optical power is extremely high, reaching tens of kW.

The reviewer’s concern refers to the peak power, which must necessarily be high to efficiently trigger nonlinear effects. However, the average optical power is on the order of a few watts, and as low as 0.4 W in the HNLf fibre used for the three-mode experiments. Given that our laser has a wall-plug efficiency of approximately 50%, this corresponds to about 0.8 W of electrical power, which could be further reduced with an optimized optical source.

Notably, this power level aligns with several commercial electronic or electromechanical optical switching devices, which typically operate in the 0.5–2 W range (see Refs. [Rev1, Rev2]). However, these devices exhibit relatively slow switching speeds—e.g., on the order of milliseconds for MEMS—and insertion losses of 1–2 dB. In contrast, our platform not only offers significantly higher switching speeds but is also virtually lossless, as the probe is switched directly within the fibre (enabling endless fibre transmission).

These numbers clearly demonstrate the practical relevance of the operations presented in this work. Furthermore, scaling these results to highly nonlinear materials, such as silicon or silicon nitride, could reduce power consumption by up to three orders of magnitude, bringing the average power down to just a few milliwatts. This highlights the enormous potential of all-optical approaches, which offer drastically lower power consumption, higher speed, and reduced losses than the electronic counterparts.

In response to this concern, we have:

(i) included the two references (Rev1 and Rev2) and three additional paragraphs discussing the abovementioned points on the average power, namely: lines 234-241: “It is worth noting that the BCB peak power [...] without compromising the capability for all-optical reconfiguration”; lines 281-283: “Similarly to the case of mode manipulation [...] the BCB average power to just a few hundred milliwatts”; lines 474-483: “Moreover, scaling these results to highly nonlinear materials [...] as the probe is switched directly within the fibre.”

(ii) specified both the peak powers and corresponding average powers throughout the manuscript to avoid any ambiguity regarding the latter.

(iii) explicitly used the label 'BCB peak power (kW)' instead of 'BCB power' for all horizontal axes in Figures 3-8."

Rev1: Coreray MEM switcher, <https://www.coreray.com/product/mems-1-n-optical-switch.html>

Rev2: Primanex optical switcher, <http://www.primanex.com.cn/en/upfile/1x2%20Maglight%20PM%20Fiber%20Optical%20Switch%20Datasheet.pdf>

4. The probe signals used are pulsed. To better illustrate the practical value of the findings, experiments using actual communication signals would have been more compelling.

Our probe is not a single pulse but a train of pulses in a return-to-zero (RZ) format, effectively emulating digital communication signals with a random sequence of bit-0 and bit-1. While our current setup is limited to 0.8 MHz due to the lack of GHz-range modulators at 1040 nm wavelength (which are significantly rarer and more expensive than those in the C-band), our simulations (see Supplementary Fig. S5) confirm that the same effects hold at several GHz, demonstrating the feasibility of operating at modern transmission rates.

Note that, beyond optical communications, our findings have practical value in other domains - an aspect we consider one of the most significant contributions of our study. As highlighted in the conclusions, tuneable all-optical modal and power splitting are crucial for nonlinear and interferometric applications, while the ability to remotely characterize the probe (Application 3) is directly relevant to remote sensing. Notably, these applications do not require high modulation speeds, and in many cases, even a single pulse is sufficient.

5. The potential for ultrahigh-speed switching, highlighted in the paper title, is certainly interesting. However, the authors provide no experimental demonstration of this capability.

The switching time, namely, the time required for the probe to transition from an initial to a final state, is determined by the duration of the BCB pulse. In our experiments, we employ sub-nanosecond BCB pulses, and indeed, as shown in Fig. 5d,e, the switching time is confirmed to be sub-nanosecond. Therefore, we do provide experimental demonstration of sub-nanosecond switching. This is why we explicitly used the term *sub-nanosecond* in the title, rather than the more generic *ultrahigh-speed*.

It is worth noting that a sub-nanosecond switching time is extremely fast compared to many traditional methods, such as MEMS, which operate on the millisecond scale (see response #3). For example, in telecom transmissions this enables switching to be completed within a time frame comparable to the bit duration, ensuring that no bits are lost. In contrast, when the switching time significantly exceeds the bit-time, multiple bits become irrecoverably lost. Moreover, as discussed in the Supplementary Information 3, switching times could plausibly reach the sub-picosecond regime with a sub-picosecond BCB pulse.

6. Moreover, the quality of the paper's presentation needs improvement. The theory section, "Probe-Control Beam Interaction" gives very limited useful information. Important details to understand the working conditions of the system—such as the power levels and the wavelength relationship between the pump and probe—are not mentioned. Additionally, the method of how to change the pump to control the matrix is not explained. Readers must frequently refer to supplementary information or the experimental results to get more information.

We agree with the reviewer's observation and have substantially revised the theory section to address these concerns. We have now incorporated in the main text some key information previously found in the 'Methods' section. We believe these revisions improve the clarity and completeness of the theory section while maintaining a balance between accessibility and technical depth.

Specifically:

(i) We have clarified that the probe and BCB are centred at the same wavelength and co-polarized. This configuration ensures automatic phase-matching of all nonlinear interactions, regardless of the fibre parameters or carrier wavelength—a cornerstone of our approach, see lines 77-90: "First, it enables physical separation [...] and probe-to-BCB frequency detuning." Note that the 'Methods' section and Supplementary Information 2 discusses the general case where the probe and BCB are not centred at the same wavelength or have different polarizations.

(ii) We have explicitly stated the power levels that characterize the probe in both linear and nonlinear regimes, which are essentially determined by the number of nonlinear lengths. This is now clearly explained in the revised text, line 99-109: "The distinction between linear and nonlinear regime [...] for the BCB based on its number of nonlinear lengths."

(iii) We have added a new section titled 'Illustration of All-Optical Reconfiguration in the Case N=2' (lines 136-178) which provides explicit analytical expressions describing the mode power distribution and core power distribution of the output probe as a function of the BCB

parameters in a multimode fibre with $N=2$ modes or multicore fibre with $N=2$ cores. This directly addresses the reviewer's question regarding "how to change the pump to control the matrix," illustrating the concept in a simple case with 2 modes and 2 cores. The general solution (for arbitrary number of modes or cores) remains detailed in the 'Methods' section, although full analytical solutions become impractical due to increased complexity.

7. In summary, while the findings in this paper are interesting, the experimental demonstrations and overall presentation quality fall short of the standards expected of Nature Communications. My main concern is the lack of significant experimental progress toward practical applications, compared to the authors' previous work.

We believe our responses have thoroughly addressed the reviewer's concerns. In summary:

(i) This work explores a fundamentally different regime than our previous study. The analogy with a single-mode fibre operated at low and high power—introduced in Response #1—illustrates this distinction clearly. Just as dynamics drastically change in that case, here, the low-power probe regime unlocks novel applications not observed in our prior work. These include fully tuneable mode and power conversion and remote sensing characterization, which were previously unattainable.

(ii) We experimentally demonstrate probe reconfiguration in fibres supporting up to six modes, showcasing the scalability of our approach beyond simple two-mode cases.

(iii) This study highlights that our platform outperforms conventional solutions in switching speed and insertion losses, while maintaining a comparable power consumption. The potential for significant power reduction via integrated photonics further reinforces its practical feasibility.

(iv) We have experimentally demonstrated sub-nanosecond switching and predict sub-picosecond switching as feasible.

(v) Beyond these results, we believe that the ability to all-optically reconfigure matrix products could serve as a powerful enabler for all-optical computing and photonic hardware for machine learning

Note that in the new version we have also introduced a new Supplementary Information 5 discussing the robustness of probe remote sensing compared to traditional techniques. This analysis further strengthens the advantages of our approach, particularly its resilience to environmental perturbations—key limitations in conventional phase-retrieval methods.

Taken together, these points underscore both the practical significance and fundamental novelty of our work.

Reviewer #2:

I have carefully reviewed the paper titled "Sub-nanosecond All-Optically Reconfigurable Photonics in Optical Fibers" and identified several critical issues that need to be addressed and revised by the authors before the manuscript can be considered for publication.

Major Issues:

1. The concept of nonlinear mode conversion in multimode optical fibers is not new and has already been explored in previous works. Notable references include:

o J. Xu, G. S. D. Gordon, T. Wilkinson, C. Peucheret, "Experimental observation of non-linear mode conversion in few-mode fiber," CLEO, 2015, paper SM2L.3.

o R.-J. Essiambre et al., "Experimental Observation of Inter-Modal Cross-Phase Modulation in Few-Mode Fibers," IEEE Photonics Technology Letters, vol. 25, no. 6, pp. 535-538, March 15, 2013.

o Dimitar I. Kroushkov, Georg Rademacher, and Klaus Petermann, "Cross mode modulation in multimode fibers," Opt. Lett., 38, 1642-1644 (2013).

The manuscript must clearly highlight the novelty of the presented work and how it advances beyond these existing studies.

We have indeed cited several studies on nonlinear mode conversion between two modes in our manuscript (Refs. [27–29]). The works suggested by the reviewer, including those by D.I. Kroushkov et al. (theoretical) and J. Xu et al. (experimental), fall into this category—describing nonlinear interactions that are fundamentally limited to only two modes. Specifically, both studies focus on mode conversion within the same mode group, e.g., between LP01x and LP01y or LP11x and LP11y.

In particular, the experimental work by J. Xu et al., which builds on the theoretical insights of Kroushkov et al., demonstrates partial mode conversion (approximately 50%) between two modes within the LP11 mode group (intragroup conversion). It does not provide a direct comparison with theoretical predictions, making it difficult to assess the precise level of control achievable.

Regarding the work by R. J. Essiambre et al., this study is representative of complex intermodal interactions but belongs to a different category. It shows that, under appropriate phase-matching conditions, a probe undergoes spectral broadening due to intermodal cross-phase modulation with a pump—an effect analogous to polarization cross-phase modulation in single-mode fibres.

We believe our work goes significantly beyond these previous studies for the following reasons:

(i) Our approach enables reconfiguration of *an arbitrary* number N of probe modes, rather than just a pair of phase-matched intragroup modes. This is demonstrated in our experiments with three-mode and six-mode fibres (see new Figure 4). The keystone of our approach is the simultaneous phase-matching of all the probe-BCB intermodal four-wave-mixing interactions, and irrespective of the fibre parameters.

(ii) Our method enables complete mode conversion (see Fig. 3a,d) rather than just partial conversion, which is the case in the above-mentioned works cited by the reviewer. And more importantly, we

demonstrate a good degree of control and predictability through agreement between theory and experiment.

(iii) We experimentally achieve probe reconfiguration on a sub-nanosecond timescale.

(iv) We demonstrate that our approach generalizes to multicore fibres, a non-trivial extension that introduces additional challenges but significantly broadens the potential impact of our results. All-optically tuneable power splitting and core-to-core sub-nanosecond switching are two key examples that we demonstrate.

(v) We introduce new applications such as robust probe remote sensing, further highlighting the versatility of our platform.

(vi) Beyond these results, the ability to reconfigure all-optically matrix \mathbf{M} enables fast reconfigurable matrix products (see Equation 1), which could serve as a powerful enabler for all-optical computing and photonic hardware for machine learning if implemented on chip.

To address the reviewer's concern and further emphasize the points mentioned above, we have now explicitly cited J. Xu et al., CLEO, 2015, paper SM2L.3. and I. Kroushkov et al. Opt. Lett., 38, 1642-1644 (2013), alongside Refs. [27–29]; and R.-J. Essiambre et al. IEEE Photonics Technology Letters, vol. 25, no. 6, pp. 535-538, March 15, 2013, alongside Refs. [10-13].

We have also added new experimental results to the main manuscript, incorporating previously supplementary data on three-mode fibres and presenting new experiments conducted in a six-mode fibre. These results are illustrated in Fig. 4, with the corresponding discussion provided in lines 225–259.

Finally, we have created a new Supplementary Information 5 that includes a discussion on the robustness of our probe remote sensing approach versus traditional techniques.

2. The manuscript does not clearly explain the underlying physics of nonlinear mode and core conversion in the backward-pumping configuration. The authors must explicitly state whether the nonlinear conversion is attributed to intermodal cross-phase modulation, four-wave mixing, or another nonlinear interaction. What specific nonlinear processes are involved? Similarly, the mechanisms behind nonlinear core conversion in multicore fibers require further clarification. Is phase matching simultaneously achieved for multiple interaction processes? More detailed explanations are essential.

We agree with the reviewer that these aspects were not sufficiently explained.

The intermodal cross-phase modulation terms between the BCB and the probe (first summation on the right-hand side of Eq. 6 in the 'Methods' section) modify the phase of each probe mode. Meanwhile, the intermodal four-wave mixing terms (second summation on the right-hand side of Eq. 6) govern the exchange of photons between probe modes, thereby enabling the redistribution of energy among them. Therefore, the modal nonlinear conversion is essentially driven by the intermodal four-wave-mixing terms.

Thanks to the configuration in use (counter-propagating with the BCB and probe at the same frequency), all the intermodal four-wave mixing terms are simultaneously phase-matched, regardless of the probe frequency or the fibre parameters. This enables energy exchange across all N probe modes, thereby allowing a complete reconfiguration of its modal state. This energy exchange is mediated by the BCB but without any net energy transfer to the probe, which therefore preserves its total energy. Note that such dynamics is not attainable in conventional co-propagating systems, where phase-matching of intermodal four wave mixing processes is achieved only between 2 modes and under specific probe-to-pump frequency detuning and/or fibre parameters.

The analysis of core-to-core nonlinear conversion in a multicore fibre is analogous to that of multimode fibres. Indeed, a mapping exists between the field distribution in each core and the modes of a multicore fibre, as described by the matrix T in Eq. (2). This matrix accounts for all linear and nonlinear phase terms, which contribute to the inter-core dynamics.

We have revised our theoretical description to explicitly highlight these key aspects, at lines 77-90: “First, it enables physical separation [...] only under strict conditions on fibre parameters and probe-to-BCB frequency detuning”, and at lines 547-552: “We note that the first summation [...] a complete reconfiguration of its modal state”.

Furthermore, in the revised version, we clarify the mechanism underlying the complete reconfiguration of the probe right from the Abstract, lines 10-12: “. This setup ensures simultaneous phase-matching of ...by tuning the control beam power”, which is reiterated in the concluding remarks, lines 442-443 “In this setup, all the probe-BCB four-wave-mixing interactions are simultaneously phase-matched, which enables a complete reconfiguration of the probe modal state”

3. The required pump power for nonlinear conversion is reported to exceed 7 kW. This raises concerns about nonlinear pulse propagation effects. The authors should discuss the implications of such high power levels in greater detail, including potential limitations and how they mitigate detrimental nonlinear effects.

We agree with the reviewer that more details should be given on this aspect.

We first note that the reported power levels in the kilowatt range refer to the peak power of the control beam, while the corresponding average optical power is significantly lower— typically < 0.5 W in the HNLF experiments and a few Watts in standard fibres.

Regarding potential detrimental nonlinear effects, we have thoroughly verified that the peak power levels and pulse width used in all experiments do not lead to undesired phenomena such as pulse distortion, spectral broadening, Brillouin scattering or wavelength conversion induced by Raman scattering or four-wave mixing. This is not surprising, given that despite the high peak power, however the fibre lengths employed are short (only 40 cm).

In response to this reviewer's concern, in the new version of the manuscript the abovementioned considerations are now discussed at lines 512-516: “Similarly, detrimental nonlinear effects, including pulse reshaping [...] Supplementary Information 4”; and at lines

609-611: “The short fibre length (40 cm) used in our experiments prevents [...] four-wave mixing-induced wavelength conversion”.

We have also created a new Supplementary Information 4, where we present typical experimental measurements of the BCB in both the time and frequency domains, at the fibre input and output, for the highest peak power levels used. These measurements confirm that no significant detrimental effects are observed, neither spectrally nor temporally.

4. The manuscript does not address how the fiber length impacts nonlinear conversion efficiency. This is an important parameter that needs to be investigated and discussed.

The role of the length is clarified in the ‘Methods’ section (System linearization: probe-BCB equations), where it appears as one of the parameters within the reconfiguration matrix **M**.

To further clarify this aspect, we have now added a new section titled ‘Illustration of All-Optical Reconfiguration in the Case N=2’ (lines 136-178), which provides analytical expressions describing the output probe in a multimode fibre with 2 modes or a multicore fibre with 2 cores. These formulas explicitly show the influence of each parameter, including the fibre length, in determining the output probe state. The general solution (for arbitrary number of modes or cores) remains detailed in the ‘Methods’ section, although full analytical solutions become impractical due to increased complexity.

5. The title should clearly specify that the work involves multimode or multicore optical fibers. Otherwise, it risks appearing redundant with the extensive body of research on all-optical processing in single-mode fibers over the past four decades.

We appreciate the reviewer's suggestion regarding the title. However, we prioritise conciseness in our title, aligning with the style commonly adopted in Nature Communications. Expanding it further might reduce its effectiveness without significantly enhancing the reader's understanding.

We note that the manuscript already makes it immediately clear that our work focuses on multimode and multicore fibres. Specifically, the first sentence of the abstract states: “*We introduce a novel all-optical platform in multimode and multicore fibres.*” This ensures that readers are aware of the specific fibre type from the outset. For these reasons, we prefer to retain the current title.

6. Minor Issues:

Figures: The resolution of the figures needs improvement to ensure clarity and readability.

We agree and we have now improved the resolution.

Terminology on Page 163: The phrase on page 163 “PBCB is coupled to a single mode, say mode 1” is unclear. Do you mean a single core ?

In that specific case, the BCB is indeed coupled to a single mode of the dual core fibre (DCF). Note that the even (odd) mode of the DCF have both core fields in-phase (anti-phase).

Reviewer #3:

In this manuscript, effort has been made towards all-optically dynamical manipulation over light within optical fiber. Besides, with a low-power probe beam and a high-power counter-propagating control beam, the article expands the application of light-light modulation from single-mode fiber (SMF) to few-mode fiber (FMF) and multi-core fiber (MCF). Experiments including mode conversion, core-to-core tunable power reconfiguration and probe remote characterization are demonstrated to underline its versatile feature and its sub-nanosecond modulation time scale. To some extent, the setups do made contributions towards reconfigurable optical networks and optical computing, I still find this work not solid enough to match the high standard of Nature Communications in terms of theory and application. Therefore, I regret to tell that I cannot recommend the publication to Nature Communications. My perspectives are as follows:

1. This manuscript does not seem to differ significantly from the previous work mentioned in the Ref [1]. The input probe light has been changed from a high-power beam that induces strong nonlinearity to a low-power beam, and lead to new dynamics. However, there is no further demonstration of a deeper exploration into the complex nonlinear multimode processes during this process.

[1] Ji, K. H. et al. Mode attraction, rejection and control in nonlinear multimode optics. *Nat. Commun.* 14, 7704 (2023).

We respectfully disagree with this assessment. We acknowledge that this might be due to our inability to clearly convey the differences between our work and Ref. [1], previously outlined in the former Supplementary Information 4. We take this opportunity to outline these distinctions.

First, we note that we have provided an analytical solution, in good agreement with experimental results, that describes the spatiotemporal dynamics of the probe in the most general case—allowing for an arbitrary number of modes and fibre parameters. This solution, entirely distinct from the theoretical results in Ref. [1], serves as strong evidence of our ability to comprehensively explore and accurately describe the multimode system in all its complexity.

Secondly, while the experimental setup and underlying equations are similar to the previous work, the difference in the probe dynamics is striking. A useful analogy can be drawn with single-mode optical fibres: given the same equations and setup, the observed dynamics drastically change depending on whether the input light is in the low-power regime (dominated by dispersion and polarization effects) or in the high-power regime (leading to phenomena such as solitons, rogue waves, or supercontinuum generation, to name a few).

Similarly, in our previous work [1], where the probe was at high-power, we observed a symmetric interaction where the probe and the BCB mutually influenced each other, leading to soliton-like (mode attraction) or anti-soliton-like (mode rejection) dynamics. Importantly, this process occurred independently of the initial probe state. In contrast, the present study reveals a fundamentally different behaviour. First, the interaction is highly asymmetric, with the BCB being only marginally affected by the probe. Second, the final state of the probe is strongly dependent on its initial condition, which underpins the novel applications proposed.

This shift in dynamics has profound implications: rather than acting as an attractor or rejector, the BCB now serves as an *all-optical modulator for the probe*, enabling on-demand probe reconfiguration directly within the fibre—a role traditionally fulfilled by external thermo-electronic modulators. More broadly, the ability to reconfigure all-optically matrix \mathbf{M} enables fast reconfigurable matrix products (see Equation 1), a functionality with profound implications for all-optical computing and photonic hardware for machine learning if implemented on chip.

In conclusion, this work fully realizes the concept of all-optical reconfiguration of the modal state. Notably, this is achieved with a low-power probe, making the proposed applications viable for real-world implementation.

To ensure these fundamental differences are fully appreciated, we have created an entirely new Section: ‘Linear vs nonlinear regime of the probe’ (line 392-438), which incorporates the aforementioned points along with a new figure (Fig.8) and the content previously reported in the former Supplementary Information 4.

2. The manuscript demonstrates the tunable conversion between modes M1 and M2, as well as core-to-core power conversion in a three-core fiber (TCF). However, these experiments do not adequately demonstrate the universality of the proposed all-optical modulation across arbitrary scenarios. In other words, the conversion involving only two modes (and the transitions among TCF) is too fundamental to showcase the superiority of this method. In my opinion, exploring conversions among a greater number of modes could be beneficial to address more complex application scenarios.

We note that in the previous version of the manuscript, we had already discussed in the former Supplementary Information 2 the results obtained in three-mode fibres. However, we recognize that presenting them in the main manuscript strengthens the clarity and impact of our findings.

In the revised version, we have introduced a new Figure 4, along with corresponding comments (line 225-259), where we present both earlier results in three-mode fibres and new experiments conducted in a six-mode fibre. Overall, we showcase several distinct cases that demonstrate our ability to manipulate the probe’s modal state on demand, even in complex scenarios.

3. I have concerns regarding the title "Sub-nanosecond all-optically reconfigurable photonics in optical fibers." The conversion time mentioned in line 192 for the multi-core fiber is not equivalent to the overall response time of the system when considering nonlinear effects. I believe this should be clearly articulated in the title to avoid ambiguity.

We have demonstrated both theoretically (Supplementary Information 3) and experimentally (Fig. 5d,e) that the conversion time, which refers to the time required for the probe to transition from its initial to final state, is determined by the duration of the BCB pulse. Specifically, in our experiments we use sub-nanosecond BCB pulses, and as shown in Fig. 5d,e, the probe reconfiguration is completed over a sub-nanosecond time scale. We are confident that this aligns with the title.

The only scenario in which nonlinear effects could impact the conversion time is if they induced temporal distortion in the BCB pulse. **However, as detailed in Supplementary Information 4—that we have added in the revised manuscript—our measurements confirm that no such distortion occurs.**

4. The formulation in line 95 is unclear. While T is described as the transmission matrix that connects the input and output ends, the wording in the manuscript may lead to confusion.

We agree with the reviewer that this part may have been unclear.

We have thoroughly revised the whole section ‘Probe-control beam interaction’, including several new details to improve clarity. Overall, we believe the updated version of this section is now clearer and more detailed, which provides a better explanation of the role of the matrix T . In addition, the section ‘Methods-System linearization’ includes explicit examples of the transformation matrix T in the multicore fibres used (lines 565-573), which further clarify its physical interpretation.

5. In Application 3, given that the transfer function is known, it is relatively straightforward to deduce the relationship with the input parameters through the far-field characteristics of the output. In my opinion, comparing the proposed approach with other conventional methods to demonstrate its superior inference accuracy would enhance the manuscript's credibility.

We thank the reviewer for this suggestion. We agree that comparing our approach with conventional methods adds further value to our work.

The conventional method is indeed based on the pre-estimation of a transmission matrix. As noted in the manuscript, when using this approach (assuming negligible mode coupling for simplicity), the input relative phase $\Delta\phi_{in}$ between two modes is estimated as $\Delta\phi_{in} = \Delta\phi_{out} - \Delta\phi_{acc}$, where $\Delta\phi_{out}$ is the relative phase measured at the fibre output, and $\Delta\phi_{acc}$ is the differential modal phase accumulated during propagation, measured through prior fibre characterization (transmission matrix). This method requires accurately estimating both $\Delta\phi_{out}$ and $\Delta\phi_{acc}$, typically necessitating complex experimental setups. However, even with highly precise phase estimations, a fundamental issue remains: the differential phase accumulation is highly sensitive to external perturbations, such as bending or temperature variations. Therefore, maintaining accurate phase estimations over time typically requires periodic calibration or active feedback mechanisms to compensate for environmental fluctuations. These approaches add further control elements, and then complexity, to the system.

In contrast, our proposed approach is entirely remote: it does not require prior measurement of $\Delta\phi_{acc}$ or $\Delta\phi_{out}$. Instead, it relies solely on the calculation of the Kerr coefficients, which we have numerically estimated, and on the measurement of modal powers at the output—quantities that are significantly easier to assess than relative phases. Furthermore, our approach is robust against external perturbations and would not require any recalibration or feedback control. This is because the

nonlinear Kerr coefficients, on which our estimation is based, are inherently stable against such perturbations.

We have revised the wording in the section ‘Application 3’ (mainly lines 354-367 and lines 377-379) to highlight these aspects.

Additionally, we have added a new Supplementary Information 5, where we present a direct comparison between our approach and the conventional transfer matrix method under varying temperature and bending conditions, emphasizing the robustness of our technique.

Reviewer #4 :

The proposed novel all-optical platform in multimode and multicore fibres in the article is interesting, but it also faces many challenges and problems:

1)The probe power is weak (linear regime), whereas the BCB is in a strongly nonlinear propagation regime. This leads to peculiar new dynamics, fundamentally different from mode attraction and rejection. The power range of the probe and BCB should be discussed to better distinguish different working dynamics (mode attraction and rejection, all-optical reconfiguration).

We had previously discussed the power ranges in the former Supplementary Information 4, where we noted that if the number of nonlinear lengths exceeds 5, the probe operates in a strongly nonlinear regime, leading to mode attraction and rejection. Conversely, if it is below 0.5, the probe remains in a linear regime, resulting in the reconfiguration dynamics described in this work.

In response to the reviewer's concern, we have:

(i) Explicitly clarified in the main manuscript the conditions that distinguish the linear and nonlinear regimes for both the probe and the BCB, which are primarily determined by the number of nonlinear lengths (dependent on power levels). This is now clearly stated in the revised text, lines 99-109: "*The distinction between linear and nonlinear regime [...] for the BCB based on its number of nonlinear lengths.*"

(ii) Introduced a new section, 'Linear vs Nonlinear Regime of the Probe,' which directly compares mode attraction/rejection with all-optical reconfiguration. This section expands upon the content previously included in the former Supplementary Information 4, including a discussion of the power range for both the probe and the BCB.

2)Although arbitrary output probe mode distribution can be achieved by tuning the BCB power in Fig. 3, we would like to know more about the purity of the pattern and how it varies with CBC.

We agree this is an important aspect to address. In the section 'Methods-Experiments', we mentioned that the reconstructed distribution pattern typically exhibits a correlation Corr as high as 99% with the measured spatial profile. In Fig. 3d-3f, we provided the output probe far-field mode distribution at three different BCB peak powers.

To better illustrate how the mode pattern changes with BCB peak power, we have provided three supplementary videos corresponding to Fig. 3d-f. These videos show the measured mode patterns at varying BCB peak powers (from 0 to the maximum) alongside reconstructed mode patterns, which are obtained from the superposition of guided modes based on mode decomposition results (relative mode power and phase). To clarify this point, we have added a statement in the section 'Application 1: Tuneable mode manipulation', see lines 207-208: "We have recorded three videos illustrating the tuneable mode manipulation dynamics in this bimodal fibre, corresponding to Fig. 3d-f (see Supplementary Videos 1-3)." We have also added a statement in the section 'Methods-Experiments', see lines 625-627: "The reconstructed mode

distributions are compared with the measurements at varying BCB peak powers for the results presented in Fig. 3d-f (see Supplementary Videos 1-3).”

3) Why was the fibre chosen to be 0.4 meter long?

As discussed in the new Section ‘Linear vs nonlinear regime of the probe’ (see response to question #1), the ability to reconfigure the probe is strongly related to the interaction length L_{int} between the probe and the BCB. The fibres’ length in our experiments ($L=0.4$ m) was chosen to satisfy $L > L_{\text{int}}$, ensuring efficient probe reconfiguration. At the same time, L remains short enough to prevent unwanted nonlinear effects that could interfere with the studied dynamics. Specifically, nonlinear effects such as Raman scattering, four-wave mixing-induced wavelength conversion, Brillouin scattering and self-phase modulation (SPM) that could temporally or spectrally distort the BCB. This distortion would, in turn, negatively impact the probe-BCB dynamics. We note that 0.4 m is a safe length, meaning that we could comfortably use up to several meters without significant issues.

In response to this reviewer’s concern, in the new version of the manuscript the abovementioned considerations are now mentioned in the ‘Method’ section, at lines 512-516: “Similarly, detrimental nonlinear effects, including pulse reshaping [...] Supplementary Information 4”; and at lines 609-611: “The short fibre length (40 cm) used in our experiments prevents [...] four-wave mixing-induced wavelength conversion”.

We have also added a new Supplementary Information 4, where we present typical experimental measurements of the BCB in both the time and frequency domains, at the fibre input and output, for the highest peak power levels used. These measurements confirm that no significant detrimental effects are observed, neither spectrally nor temporally, over the short fibres span considered.

4) A single 0.5 ns BCB pulse shifts the core-to-core power ratio at the DCF output from 35/65, when the BCB is 192 off, to 65/35 when the BCB peak power is 5 kW. This relative change is only 30%, how can it reach the application level of full light control, and the CBC power is also particularly high, so the application prospects seem very limited.

The result shown in Fig. 5d-e is an example showing core-to-core probe power swapping, where the 2 cores mutually exchange their power (i.e. the power ratio shifts from 35/65 to 65/35). This result is intended to illustrate the sub-nanosecond transition of the probe state.

However, as seen in Fig. 5a-c, the control over the probe dynamics is actually much broader. In Fig. 5a (power splitting), we achieve any splitting ratio from 100/0 to 50/50 by tuning the BCB peak power from 0 to 9 kW, fully covering the operational range of an optical power splitter. In Fig. 5b (power combining), we start with power distributed between both cores—35/65 in our example—and combine it into a single core. This is achieved remarkably well, as we manage to combine nearly all the power into one core, reaching a ratio of 92/8 at a BCB peak power of 11 kW. Almost full 100/0 combination could be attained at ~14 kW peak BCB power (not available in our setup). Similarly, in Fig. 5c (core-to-core switching), the same principle applies.

Regarding the power used, it should be noted that the average optical power is on the order of a few watts, and is reduced to just ~ 0.4 W in highly nonlinear fibres (**see the new results in three and six modes fibres in Figure 4 and related discussion at lines 225-259**). Given that our laser has a wall-plug efficiency of approximately 50%, this corresponds to about 0.8 W of electrical power, which could be further reduced with an optimized optical source.

Notably, this power level aligns with several commercial electronic or electromechanical optical switching devices, which typically operate in the 0.5–2 W range (see Refs. [Rev1, Rev2]). However, these devices exhibit relatively slow switching speeds—e.g., on the order of milliseconds for MEMS—and insertion losses of 1–2 dB. In contrast, our platform not only offers significantly higher switching speeds but is also virtually lossless, as the probe is switched directly within the fibre (enabling endless fibre transmission). These figures clearly demonstrate the practical relevance of the operations presented in this work. Furthermore, scaling these results to highly nonlinear materials, such as silicon or silicon nitride, could reduce power consumption by up to three orders of magnitude, bringing the average power down to just a few milliwatts. This highlights the enormous potential of all-optical approaches, leading to drastically lower power consumption, higher speed, and reduced losses than the electronic counterparts.

In response to this observation, we have:

(i) included the two references (Rev1 and Rev2) and three additional paragraphs discussing the abovementioned points on the average power, namely: lines 234-241: “It is worth noting that the BCB peak power [...] without compromising the capability for all-optical reconfiguration”; lines 278-283: “By introducing the BCB and tuning its peak power [...] the BCB average power to just a few hundred milliwatts”; lines 474-483: “Moreover, scaling these results to highly nonlinear materials [...] as the probe is switched directly within the fibre.”

(ii) Specified both the peak powers and corresponding average powers throughout the manuscript to avoid any ambiguity regarding the latter.

(iii) Explicitly used the label 'BCB peak power (kW)' instead of 'BCB power' for all horizontal axes in Figures 3-8."

(iv) Clarified that the instance shown in Fig.5d-e is just an example to illustrate core-to-core power swapping at the sub-nanosecond time scale, see line 293-294: “Fig. 5d,e illustrate an example of core-to-core power swapping at the sub-nanosecond time scale”;

(v) Further strengthened the argument that our control over the probe dynamics is very broad, line 279-280: “we achieved any arbitrary power ratio $X/(100-X)$ [...] as required for a fully tunable optical power splitter”

Rev1: Coreray MEM switcher, <https://www.coreray.com/product/mems-1-n-optical-switch.html>

Rev2: Primanex optical switcher, <http://www.primanex.com.cn/en/upfile/1x2%20Maglight%20PM%20Fiber%20Optical%20Switch%20Datasheet.pdf>

5)The larger error in panel a of Fig. 6 is due to the large power imbalance among the two input probe modes (92% and 8%, respectively). This means that if the power difference is too large, the error will be very large. Can this power range be preliminarily defined?

In our experiments, we do have control over the power imbalance between the input probe modes. However, the goal of Fig. 7 is to demonstrate that we can reconstruct the relative phase from remote (having access only to the output probe) regardless of the input probe state. In a real remote sensing scenario, it would therefore not be possible to preliminarily define the power imbalance of the input probe modes, as we would not have direct access to it.

It is worth noting that even in the presence of a significant power imbalance in Fig. 7a, our phase estimation remains accurate, with an error of 0.24 rad (13.75°). Note also that any technique relying solely on the intensity pattern faces the same challenge. That is, when the power imbalance between the modes is high, the dominant mode almost entirely determines the total intensity, while the weaker mode has minimal influence. As a result, variations in the relative phase have little to no effect on the overall intensity, making the estimation of the relative phase inherently difficult.

6)The most fatal problem with the method proposed in the article is that optical fibers are particularly sensitive to changes in the external environment. How to overcome this challenge should be discussed.

From a general perspective, it is important to note that for short fibres like those used, the issue of external environmental variations is significantly reduced compared to telecommunications fibres, which can span tens of kilometres.

Furthermore, the impact of these changes and potential solutions depends on the specific application. For instance, when implementing a tuneable mode converter (Fig. 3), or a tuneable power splitter (Fig. 5a), power combiner (Fig. 5b), or core-to-core switcher (Fig.5c and Fig.6), an obvious solution would be to block the fibre in place and enclose it in a protective casing to shield it from mechanical shocks and thermal fluctuations. This approach is commonly employed in commercial fibre amplifiers and fibre optical sources.

In the case of remote sensing, as shown in Fig. 7, the situation could be more critical, as it may require a deployable and loose optical fibre not enclosed in a protective box. However, as demonstrated in the new Supplementary Information 5, our remote sensing approach proves to be very robust against external perturbations. This is because it is insensitive to phase variations accumulated during propagation, which are typically highly susceptible to external disturbances.

In the updated version of the manuscript, we have added a new Supplementary Information 5 where we discuss the robustness of our remote sensing approach against external perturbations. We have also revised the wording in the section ‘Application 3’ (manly lines 351-365 and lines 377-379) to highlight these aspects.

Response to reviewers' comments:

Reviewer #2 (Remarks to the Author):

I have carefully reviewed the authors' responses to the referees' comments and analyzed the revised manuscript in detail. The authors have addressed most of my initial concerns, and the conclusions are now better supported by additional data and expanded discussion. While some weaknesses remain, I believe the manuscript has improved significantly and is suitable for publication in Nature Communications.

We sincerely thank the reviewer for the thoughtful re-evaluation of our manuscript. We are pleased to hear that the revised manuscript is considered significantly improved and now suitable for publication. We also acknowledge that some weaknesses remain and will continue to keep these in mind for future studies. We greatly appreciate the reviewer's constructive feedback, which has helped us enhance the quality and clarity of our work.

Reviewer #3 (Remarks to the Author):

The authors have revised the manuscript, and addressed the issues I have raised before. Now I can accept it.

We thank the reviewer for reconsidering our manuscript. We appreciate your acknowledgement that the revisions have addressed your previous concerns, and we are grateful for your support of our work.

Reviewer #4 (Remarks to the Author):

The author's revisions and responses are both very objective, reinforcing the noteworthy results. It represents a paradigm shift in all-optical control, moving beyond conventional electro-optical methods. The combination of speed (sub-ns), energy efficiency (mW-level in highly nonlinear fibers), and multi-functionality (switching/splitting/remote sensing) aligns perfectly with next-gen photonics needs. But I hope the authors can provide a limit prediction, which is the maximum number of modes or fiber cores that this method can handle simultaneously, although new experiments conducted in a six-mode fibre was presented. By the way, the title of the article is a little bit too grand, as essentially referring to the manipulation of light

We thank the reviewer for the encouraging assessment of our work. We are particularly pleased that the reviewer recognizes the paradigm shift our approach offers in the context of all-optical control.

Regarding the reviewer's insightful suggestion to include a limit prediction on the maximum number of modes or fibre cores that this method can handle simultaneously, we agree that this is an important consideration. Our experimental results (with number of modes up to 6) are supported by

a theoretical model that aligns with the experimental findings and extends to multimode and multicore fibres with an arbitrary number of modes and cores. However, in practical experiments, the maximum number of modes or cores is constrained by several factors, including the total available control beam power (e.g., laser peak power), coupling efficiency across different fibre modes or cores, and the nonlinear response of the fibre material. These parameters collectively limit the extent to which the probe signal can be reconfigured.

As such, it is challenging to provide a definitive upper limit. Nevertheless, we note that scalability can be improved by employing highly nonlinear materials, increasing the available laser power, and optimizing coupling strategies—such as using mode-division multiplexers—to more effectively excite and manage a larger number of modes or cores.

We also appreciate the comment on the title. After careful consideration, we have chosen to retain the current title, as we believe it reflects the broader implications of our work in enabling versatile and fully optical manipulation of light. However, we respect the reviewer's viewpoint and have ensured that the abstract and introduction clearly define the scope of the study to avoid overgeneralization.

We thank the reviewer again for the valuable feedback and support.